# Is uniform expressivity too restrictive?
# Towards efficient expressivity of GNNs*

**Sammy Khalife**[†]
khalife.sammy@cornell.edu

**Josué Tonelli-Cueto**[‡]
josue.tonelli.cueto@bizkaia.eu

## Abstract

Uniform expressivity guarantees that a Graph Neural Network (GNN) can express a query without the parameters depending on the size of the input graphs. This property is desirable in applications in order to have a number of trainable parameters that is independent of the size of the input graphs. Uniform expressivity of the two variable guarded fragment (GC2) of first order logic is a well-celebrated result for Rectified Linear Unit (ReLU) GNNs Barceló et al. (2020). In this article, we prove that uniform expressivity of GC2 queries is not possible for GNNs with a wide class of Pfaffian activation functions (including the sigmoid and tanh), answering a question formulated by Grohe (2021). We also show that despite these limitations, many of those GNNs can still efficiently express GC2 queries in a way that the number of parameters remains logarithmic on the maximal degree of the input graphs. Furthermore, we demonstrate that a log-log dependency on the degree is achievable for a certain choice of activation function. This shows that uniform expressivity can be successfully relaxed by covering large graphs appearing in practical applications. Our experiments illustrates that our theoretical estimates hold in practice.

## 1 Introduction

Graph Neural Networks (GNNs) form a powerful computational framework for machine learning on graphs, and have proven to be very performant methods for various applications ranging from analysis of social networks, structure and functionality of molecules in chemistry and biological applications Duvenaud et al. (2015); Zitnik et al. (2018); Stokes et al. (2020); Khalife et al. (2021), computer vision Defferrard et al. (2016), simulations of physical systems Battaglia et al. (2016); Sanchez-Gonzalez et al. (2020), and techniques to enhance optimization algorithms Khalil et al. (2017); Cappart et al. (2021) to name a few. Significant progress has been made in recent years to understand their computational capabilities, in particular regarding functions one can compute or express via GNNs. This question is of fundamental importance as it precedes any learning aspect, by asking a description the class of functions one *can hope* to learn via a GNN. A better understanding of the dependency of the expressivity on the architecture of the GNN (activation and aggregation functions, number of layers, inner dimensions, ...) can also help to select the most appropriate one, given some basic knowledge on the structure of the problem at hand. One common approach in this line of research is to compare standard algorithms on graphs to GNNs. For example, if a given algorithm is able to distinguish two graph structures, is there a GNN able to distinguish them (and conversely)? The canonical algorithm that serve as a basis of comparison is the *color refinement* algorithm (closely related to *Weisfeler-Lehman* algorithm), a heuristic that almost solve graph isomorphism Cai et al. (1992). It is for example well-known that the color refinement algorithm refine the standard GNNs, and that the activation function has an impact on the the ability of a GNN to refine color refinement Morris et al. (2019); Grohe (2021); Khalife & Basu (2023).

---

[*]Both authors contributed equally. Order is alphabetical.
[†]School of Operations Research and Information Engineering, Cornell Tech, Cornell University, NY, USA
[‡]Department of Applied Mathematics and Statistics, Johns Hopkins University, Baltimore, USA

The expressivity of GNNs can also be compared to queries of an appropriate logic. A first form of comparison relies on *uniform expressivity*, where the size of the GNN is not allowed to grow with the input graphs. Given this assumption, a first basic question is to determine the fragment of logic that can be expressed via GNNs, in order to describe queries that one can possibly learn via GNNs. For instance with first-order-logic, one can express the query that a vertex of a graph is part of a triangle. It turns out standard aggregation-combine GNNs cannot express that query, and more generally, a significant portion of the first order logic is removed. The seminal result of Barceló et al. (2020) states that aggregation-combine GNNs with sum aggregations ($\Sigma$-AC-GNNs) allow to express at best a fragment of the first order logic (called graded model logic, or GC2), and can be achieved with Rectified Linear Unit (ReLU) activations. Conversely, ReLU $\Sigma$-AC-GNNs can express uniformly queries of GC2. In Grohe (2023), the author presents more general results about expressivity in the uniform setting, introducing a new logical guarded fragment (GFO+C) of the first order with counting (FO+C), in order to describe the expressivity of the general message passing GNNs, when no assumption is made on the activation or aggregation function. The author shows that GC2 $\subseteq$ GNN $\subseteq$ GFO+C, where both inclusions are strict. See Figure 1 for a representation of the known result for uniform expressivity.

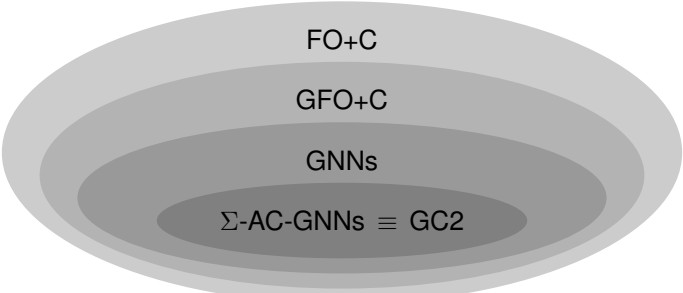

Figure 1: Main logical classes to describe the uniform expressivity of GNNs. If no restriction is made on the activation and aggregation functions, standard message passing GNNs can express slightly more than GC2 queries, but less than GFO+C queries. The sum-aggregation aggregation combine GNNs ($\Sigma$-AC-GNNs) exactly express GC2 queries.

Evidently, the logic of GNNs depends on the choice of architecture. Recent work include the impact of the activation function aggregation function on uniform expressivity Khalife (2023); Rosenbluth et al. (2023). In Grohe & Rosenbluth (2024), the authors also study the expressivity depending on the type of messages that is propagated (message that depends only on the source vs vs. dependency on source *and* target vertex). They show that in a non-uniform setting, the two types of GNNs have the same expressivity, but the second variant is more expressive uniformly. Additionally, it was also showed that GNNs with rational activations and aggregations do not allow to express such queries uniformly, even on trees of depth two Khalife (2023). More generally, a complete characterization of the activation function that enables GC2 remains elusive.

By allowing the size of the GNN to grow with the input graphs, it is possible to relax the uniform notion of expressivity and ask analogous questions to the uniform case. For instance, what is the portion of first-order logic one can express over graphs of bounded order with a GNN? What is the required size of a GNN with a given activation to do so? In this setup, it becomes now possible that general GNNs express a *broader* class than the uniform one (GFO+C), and that $\Sigma$-AC-GNNs express a broader class than GC2. Such question has been explored in the work of Grohe (2023), that describes the new corresponding fragments of FO+C. In particular, the author proves an equivalence between a superclass of GFO+C, called GFO+C$_{nu}$ and general GNNs with *rational piecewise linear* (rpl)-approximable functions. More precisely, a query can be expressed by a family of rpl-approximable GNNs whose number of parameters grow polynomially with respect to the input graphs, if and only if the query belongs to GFO+C$_{nu}$, a more expressive fragment than GFO+C. The GNNs used to prove this equivalence only requires linearized sigmoid activations ($x \mapsto \min(1, \max(0, x))$) and $\Sigma$-aggregation. Similarly to the uniform case, the set of activation functions that allows maximal expressivity still needs to be better understood. If one restricts to $\Sigma$-AC-GNNs and the well-known GC2 fragment, one can ask for example, given a family of activation functions, how well $\Sigma$-AC-GNNs with those activation functions can express GC2 queries. For example, given an activation function,

and integer $n$, what is the number of parameters of a $\Sigma$-AC-GNNs needed to express GC2 queries of depth[1] $d$ over graphs of order at most $n$?

**Main contributions.** Our first contribution goes towards a more complete understanding of both uniform and non-uniform expressivity of GNNs, with a focus on $\Sigma$-AC-GNNs. We first show that similarly to rational GNNs, there exist a large family of GC2 that cannot be expressed with $\Sigma$-AC-GNNs with bounded Pfaffian activations including the sigmoidal activation $x \mapsto \frac{1}{1+\exp(-x)}$, function that was so far conjectured to be as expressive as the ReLU $x \mapsto \max(0, x)$ units in this setup Grohe (2021). More precisely, we prove that GNNs within that class cannot express all GC2 queries uniformly, in contrast with the same ones replaced with ReLU activations. Our second contribution adds on descriptive complexity results Grohe (2023) by providing new complexity upper-bounds in the non-uniform case for a class of functions containing the Pffafian ones. Our constructive approach takes advantage of the composition power, providing some simple assumptions on the activation functions, endowing deep neural networks to approximate the step function $x \mapsto \mathbf{1}_{\{x>0\}}$ efficiently. Our experiments confirm our theoretical statements on synthetic datasets.

The rest of this article is organized as follows. Section 2 presents the basic definitions of GNNs, Pfaffian functions and the background logic. In Section 3, we state our main theoretical results, in comparison with the existing ones. Section 4 presents an overview of the proof of our negative result that builds on these properties. Section 5 presents the core ideas to obtain our non-uniform expressivity upper-bounds. Technical details including additional definitions and lemmas are left in the appendix. In Section 6, we present the numerical experiments supporting our theoretical results. We conclude with some remarks, present the limitations of our work in Section 7.

## 2 PRELIMINARIES

**Notations.** In the following, for any positive integer $n$, $[n]$ refers to the set $\{1, \cdots, n\}$. We assume the input graphs of GNNs to be finite, undirected, simple, and vertex-colored with $\ell \geq 1$ colors: a graph is a tuple $G = (V(G), E, \lambda)$ consisting of a finite vertex set $V(G)$, a binary edge relation $E \subset V(G)^2$ that is symmetric and irreflexive, representing the edges; and a map $\lambda : V(G) \to [\ell]$, representing the colors (or labels). The number of colors $\ell$ is fixed and does not depend on $n$. In $G$, $\mathcal{N}_G(v)$ will denote the set of vertices adjacent to $v$ in $G$ (not including $v$). Given a function $\sigma : \mathbb{R} \to \mathbb{R}$, $\sigma'$ will denote the derivative and $\sigma^N = \sigma \circ \cdots \circ \sigma$ (i.e., $\sigma^1 = \sigma$ and $\sigma^N = \sigma^{N-1} \circ \sigma$) the $N$-th iterated composition of $\sigma$. The function $\log$ refers to the logarithm in base 2.

**Definition 2.1** (Neural network). *A $(k+1)$-layer neural network (NN) with activation function $\sigma : \mathbb{R} \to \mathbb{R}$ with input dimension $w_0$ and output dimension $w_{k+1}$ is formed by a sequence of $k+1$ affine transformations*

$$T_i : \mathbb{R}^{w_i} \to \mathbb{R}^{w_{i+1}} \qquad (i \in \{0, \ldots, k\})$$

*where $w_0, \ldots, w_{k+1}$ are positive integers. The function that the NN computes is the function $f : \mathbb{R}^{w_0} \to \mathbb{R}^{w_{k+1}}$ given by*

$$f = T_k \circ \sigma \circ T_{k-1} \circ \cdots T_1 \circ \sigma \circ T_0$$

*where $\sigma$ is applied pointwise for every coordinate of each inner layer's output. The size of the neural network is $w_0 + w_1 + \cdots + w_k + w_{k+1}$.*

**Definition 2.2** (Graph Neural Network (GNN)). *A GNN[2] is a recursive embedding of vertices of a graph. At each iteration, the GNN combines the information at the current vertex as well as the aggregated information on the vertex's neighborhood using a neural network. More precisely, a GNN with $T$ iterations, activation function $\sigma : \mathbb{R} \longrightarrow \mathbb{R}$ and input size $\ell$ is a sequence of $T$ combination functions*

$$\mathsf{comb}_t : \mathbb{R}^{d_t} \times \mathbb{R}^{d_t} \to \mathbb{R}^{d_{t+1}} \qquad (t \in \{0, \ldots, T-1\})$$

*that are NNs with activation function $\sigma$, and where $d_0 = \ell$. Given a graph $G$ with colors $1, \ldots, \ell$, the GNN builds $T+1$ vertex embeddings $\xi^t(v, G) \in \mathbb{R}^{d_t}$ ($t \in \{0, \ldots, T-1\}, v \in V(G)$) as follows:*

---

[1]The notion of depth will be defined more precisely in Section 2. For the time being, one can think of the depth as a measure of complexity measure on the space of queries.

[2]This definition coincides with sum-aggregation-combine GNNs ($\Sigma$-AC-GNNs) in the literature. In the following, for simplicity, unless state otherwise, we refer to $\Sigma$-AC-GNNs simply as GNNs. Some of our results generalize to a slightly more general class of aggregation functions, see Remark B.1 in the appendix.

○ The initial embedding $\xi^0(v, G)$ is given by $\xi^0(v, G) = e_{\lambda(v)}$ where $e_i \in \mathbb{R}^\ell$ is the $i$th canonical vector and $\lambda(v) \in [\ell]$ the label of $v$.

○ At iteration $t$, $\xi^{t+1}(v, G)$ is computed from $\xi^t(v, G)$ via the update rule

$$\xi^{t+1}(v, G) = \mathsf{comb}_t\left(\xi^t(v, G), \sum_{w \in \mathcal{N}_G(v)} \xi^t(w, G)\right). \tag{1}$$

The *size* of the GNN is the maximum size of the NNs given by the combination functions.

In the context of this paper, we will focus on the following activation functions:

- **ReLU:** The *Rectified Linear Unit* $\mathrm{ReLU} : \mathbb{R} \to \mathbb{R}_{\geq 0}$ is defined as $\mathrm{ReLU}(x) = \max\{0, x\}$.
- **CReLU:** The *Clipped Rectified Linear Unit* $\mathrm{CReLU} : \mathbb{R} \to [0, 1]$ defined by $\mathrm{CReLU}(x) = \min\{\max\{0, x\}, 1\}$.
- **tanh:** The *Hyperbolic Tangent* $\tanh : \mathbb{R} \to (-1, 1)$ given by $\tanh(x) = \frac{e^x - e^{-x}}{e^x + e^{-x}}$.
- **Sigmoid:** The *Sigmoid* $\mathrm{Sigmoid} : \mathbb{R} \to (0, 1)$ defined by $\mathrm{Sigmoid}(x) = \frac{1}{1 + e^{-x}}$.

**Definition 2.3** (GC2 (Barceló et al., 2020; Grohe, 2021)). A *GC2 query* is a formula with one free variable obtained from the always true empty formula $\top$ and atomic formulas $\mathsf{Col}(x)$ (returning 1 or 0 for one of the palette colors) through one of the following operations

$$\neg\phi(x), \quad \phi(x) \wedge \psi(x) \quad \text{and} \quad \exists^{\geq N} y(E(x, y) \wedge \phi(y))$$

where $\neg$ is the logical negation, $\wedge$ the logical conjunction, $\exists^{\geq N}$, with $N$ a positive integer, means "there exist at least $N$", and $E(x, y)$ means "$x$ and $y$ are adjacent in the considered graph". The *depth*, $\mathrm{D}(\cdot)$ is a complexity measure of GC2 queries defined recursively as follows: for the always true empty formula $\top$, $\mathrm{D}(\top) = 0$; for any atomic formula, $\mathrm{D}(\mathsf{Col}(\cdot)) = 1$; and for the non-atomic formulas, $\mathrm{D}(\neg\phi) = \mathrm{D}(\phi) + 1$, $\mathrm{D}(\phi \wedge \psi) = \mathrm{D}(\phi) + \mathrm{D}(\psi) + 1$ and $\mathrm{D}(\exists^{\geq N} y(E(x, y) \wedge \phi(y))) = \mathrm{D}(\phi) + 1$.

*Remark* 2.1. GC2 queries are formulas from the so-called *guarded model logic* (GC), but restricted with two variables. In general, GC formulas are free formulas constructed using Boolean connectives, such as $\neg$ and $\wedge$, and quantifiers that range over the neighbors of the current node, such as $\exists^{\geq N} y(E(x, y) \wedge \phi(y))$.

*Remark* 2.2. Given our definition, the depth of a query depends on its writing. For example, if $\phi$ is a GC2 query, $\phi$ and $\phi \wedge \phi$ express the same GC2 query, but the latter has larger depth.

**Definition 2.4.** Given a graph $G$, a vertex $v$ and a unary query $Q$, $Q(v, G) = 1$ if $Q(v)$ is true in $G$ and $Q(v, G) = 0$ if $Q(v)$ is false in $G$.

*Example* 2.1 (Barceló et al. (2020)). All GC2 formulas define unary queries. Suppose $\ell = 2$ (number of colors), and for illustration purposes $\mathsf{Col}_1 = \mathrm{Red}$, $\mathsf{Col}_2 = \mathrm{Blue}$. Then

$$\gamma(x) := \mathrm{Blue}(x) \wedge \exists y\left(E(x, y) \wedge \exists^{\geq 2} x\left(E(y, x) \wedge \mathrm{Red}(x)\right)\right)$$

queries if $x$ is blue and it has at least one neighbor with two red neighbors. Then $\gamma$ is in GC2. Now,

$$\delta(x) := \mathrm{Blue}(x) \wedge \exists y\left(\neg E(x, y) \wedge \exists^{\geq 2} x\left(E(y, x) \wedge \mathrm{Red}(x)\right)\right)$$

is not in GC2 because the use of the guard $\neg E(x, y)$ is not allowed. However,

$$\eta(x) := \neg\left(\exists y\left(E(x, y) \wedge \exists^{\geq 2} x\left(E(y, x) \wedge \mathrm{Blue}(x)\right)\right)\right)$$

is in GC2 because the negation $\neg$ is applied to a formula in GC2.

**Definition 2.5.** (Khovanskiĭ, 1991) Let $U \subseteq \mathbb{R}^n$ be an open set. A Pfaffian function $\sigma : U \to \mathbb{R}$ is an analytic function for which there is $\alpha \geq 1$ and a chain of analytic real functions $f_1, \cdots, f_r : U \to \mathbb{R}$ such that $f_r = \sigma$ and that satisfies the following differential equations

$$\partial f_i / \partial x_j = P_{i,j}(x, f_1(x), \ldots, f_i(x)))$$

where $P_{i,j} \in \mathbb{R}[x_1, ..., x_n, y_1, ..., y_i]$ are polynomials on $(n + i)$ variables of degree at most $\alpha$.

**Proposition 2.1.** $\tanh$ *and* $\mathrm{Sigmoid}$ *are bounded Pfaffian functions.*

## 3 STATEMENT OF THE CONTRIBUTIONS

Our contributions focus on the expressivity of GC2 queries by GNNs.

**Definition 3.1.** Given a unary query $Q$ and a GNN computing a vertex embedding $\xi$, we say that the *GNN expresses $Q$ over a set of graphs $\mathcal{X}$* if there is a real $\epsilon < 1/2$ such that for all graphs $G \in \mathcal{X}$ and vertices $v \in V(G)$,

$$\begin{cases} \xi(v, G) \geq 1 - \epsilon & \text{if } Q(v, G) = 1 \text{ and} \\ \xi(v, G) \leq \epsilon & \text{if } Q(v, G) = 0 \end{cases}$$

We say that the GNN expresses *uniformly $Q$* if $\mathcal{X}$ is the set of all graphs.

The following result states the best known upper-bound on the number of iterations and size of GNNs with activation function CReLU to express GC2 uniformly.

**Theorem 3.1.** *(Barceló et al., 2020; Grohe, 2021) For every GC2 query $Q$ of depth $d$, there is a GNN with activation function CReLU, $d$ iterations and size at most $4d$ that express $Q$ over all graphs. Conversely, any GNN with activation function CReLU expresses a GC2 query.*

*Remark* 3.1. We can further restrict the GNN in Theorem 3.1 by requiring its combination functions to be of the form $\text{comb}_t(x, y) = \text{CReLU}(Ax + By + C)$ with $A \in \{-1, 0, 1\}^{d \times d}$, $B \in \{-1, 0, 1\}^{d \times d}$ and $C \in \mathbb{Z}^d$ independent of $t$.

*Remark* 3.2. Theorem 3.1 admits the a partial converse for beyond AC-GNNs Barceló et al. (2020): if a unary query is expressible by a GNN and also expressible in first-order logic, then it is expressible in GC2.

Until now, the question about uniform expressivity has been mainly open for other activation functions such as $\tanh$ and Sigmoid. Our contributions is two-fold. On the one hand, we show that uniform expressivity is not possible. On the other hand, we show that, despite this impossibility, a form of almost-uniform expressivity is achievable.

We briefly overview these two contributions below, giving more details in Sections 4 and 5.

### 3.1 IMPOSSIBILITY OF UNIFORM EXPRESSIVITY

**Definition 3.2** (Superpolynomial)**.** Let $f : \mathbb{R} \to \mathbb{R}$ be a function having a finite limit $\ell \in \mathbb{R}$ in $+\infty$. We say that *$f$ converges superpolynomially* to $\ell$ iff for every polynomial $P \in \mathbb{R}[X]$, $\lim_{x \to +\infty} P(x)(f(x) - \ell) = 0$.

We say that a Pfaffian function is *superpolynomial* if it is non-constant and verifies this condition.

Our first result extends the negative result of rational GNNs to the class of bounded superpolynomial Pfaffian GNNs.

**Theorem 3.2.** *Let $\sigma : \mathbb{R} \to \mathbb{R}$ be a bounded superpolynomial Pfaffian activation function. Let $\xi$ be the output of a GNN with activation function $\sigma$. Then there is a GC2 query $Q$, such that for any $\epsilon > 0$ there exist a pair of rooted trees $(T, T')$ of depth $2$ with roots $(s, s')$ such that*

    *i) $Q(s, T) = 1$ and $Q(s', T') = 0$*

    *ii) $|\xi(s', T') - \xi(s, T)| < \epsilon$.*

**Corollary 3.3.** *Let $\sigma \in \{\text{Sigmoid}, \tanh\}$. There exists a GC2 query $Q$ such that no GNN with activation $\sigma$ can express $Q$ uniformly over all graphs.*

We want to emphasize that the above query does not depend on the choice of the GNN provided it has this type of activation function.

### 3.2 ALMOST-UNIFORM EXPRESSIVITY

Our second contribution shows that for a large class of activation functionss, which we call step-like activation function (see Definition 5.1 in Section 5 below) we have almost-uniform expressivity of GC2 queries.

**Theorem 3.4.** *Let $\sigma$ be a step-like activation function. Then, for every GC2 query $Q$ of depth $d$ and every positive integer $\Delta$, there is a GNN with activation function $\sigma$, $d$ iterations and size $O(d \log \Delta)$ that express $Q$ over all graphs with degree $\leq \Delta$. Moreover, there exists a family of step-like activation functions such that the size of the GNN can be further reduced to $O(d \log \log \Delta)$.*

**Corollary 3.5.** *Let $\sigma \in \{\text{Sigmoid}, \tanh\}$. Then, for every GC2 query $Q$ of depth $d$ and every positive integer $\Delta$, there is a GNN with activation function $\sigma$, $d$ iterations and size $O(d \log \log \Delta)$ that express $Q$ over all graphs with degree $\leq \Delta$.*

We want to emphasize that the function $\log \log \Delta$ grows so slowly that for almost all theoretical and practical purposes we can consider it to be a constant function, hence our usage of the expression "almost-uniform expressivity".

## 4 SUPERPOLYNOMIAL BOUNDED PFAFFIAN GNNS: IMPOSSIBILITY

Let us overview why Theorem 3.2 holds, expanding all details in Appendix B. Our first result makes use of specific families of input trees that allow us to *saturate* the output signal of a GNN by increasing appropriately their number of vertices in certain regions. The input trees, pictured in Figure 2 are carefully chosen in order to return distinct outputs to query of GC2. The gist of our proof consists in showing that by increasing the order of the trees in a suitable manner, the output signals of the GNNs of the considered class can be made arbitrarily close. This is achieved by proving a stronger property on all vertices on the trees stated below. To do so, we first need the intermediate definition:

$$S_r := \{\phi : \mathbb{R}^{1+(r-1)} \to \mathbb{R} : \forall P \in \mathbb{R}[X], \forall x \in \mathbb{R} \lim_{y \to +\infty} |P(y_{r-1})\phi(x, y_1, \cdots, y_{r-1})| = 0\}$$

where for vectors $z \in \mathbb{R}^d$, and functions $F : \mathbb{R}^d \to \mathbb{R}$, $\lim_{z \to +\infty} F(z) = 0$ means that for every $\epsilon > 0$ there exists $\beta \geq 0$, such that $\min(z_1, \cdots, z_d) \geq \beta$ implies $|F(z)| < \epsilon$.

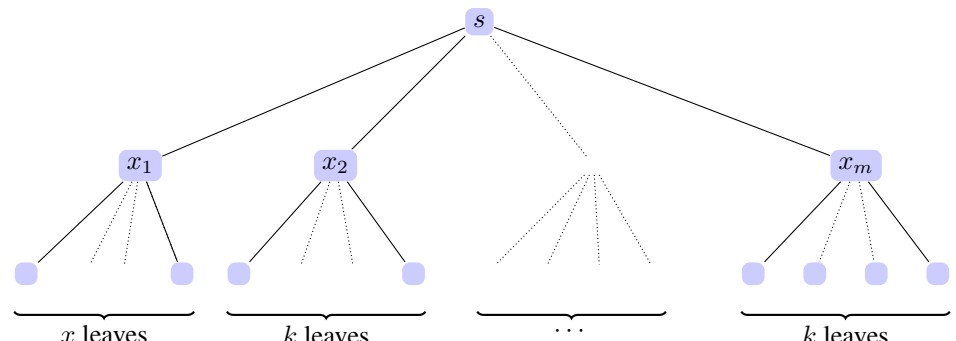

Figure 2: Tree $T[x, k, m]$ with root $s$

For non-negative integers $k$ and $m$, let $T[x, k, m]$ be the rooted tree of Figure 2. In this tree, the root $s$ has $m$ descendants $x_i$, having $x_i$ $x$ descendants for $i = 1$ and $k$ descendants for $i \geq 2$. For ease of notation, whenever context allows, we will refer to $T[x, k, m]$ simply as $T$ and to any of the descendants of $x_i$ as $l_i$. For any non-negative integer $t$ and $v \in V(T)$, let $\xi^t(v, T)$ be the embedding returned for node $v$ by a GNN with bounded superpolynomial Pfaffian activation function $\sigma$ after $t$ iterations. Let $M > 0$ be an integer. We will prove by induction on $t$ that for any $t \in \{1, \cdots, M\}$:

$$\xi^t(s, T) = v_t + \eta_t(x, k, m)$$
$$\xi^t(x_1, T) = g_t(k) + \epsilon_t^1(x, k, m) \qquad \xi^t(x_{i \geq 2}, T) = g_t(k) + \epsilon_t(x, k, m)$$
$$\xi^t(l_1, T) = h_t(k) + \nu_t^1(x, k, m) \qquad \xi^t(l_{i \geq 2}, T) = h_t(k) + \nu_t(x, k, m)$$

with the following properties:

  i) $v_t \in \mathbb{R}^{d_t}$ is constant with respect to $x$, $k$ and $m$.

  ii) Each coordinate of $g_t$ and $h_t$ are bounded Pfaffian functions of $k$ that depend only on the combination and aggregation functions and the iteration $t$.

iii) Each coordinate of the functions $\nu_t, \eta_t, \epsilon_t, \epsilon_t^1, \nu_t^1$ are in $S_3$.

A few comments are in order. Recall that $S_3$ is the set of functions $\mathbb{R}^3 \to \mathbb{R}$ such that, for any fixed first argument, the function of the second and third argument tend (collectively) *superpolynomially* to 0 with respect to the third argument. This asymptotic behavior will arise from our assumptions made on the Pfaffian activation functions, and will be used in our inductive argument. One immediate corollary is that we can decompose the signal at the root vertex into a main component $v_t$ and an error term that tends to 0 when the order of the trees to $+\infty$. The main signal $v_t$ will depend *only* on the number of iterations, not on $x$, $m$ or $k$. In the other vertices, the main component will only be a function of $k$. This will prove for instance, that the the two outputs of the GNN on the pair $(T[0, k, m], T[1, k, m])_{(k,m) \in \mathbb{N}^2}$ converge to the same limit, when $k$ and $m$ tend to $+\infty$. However, for every $k, m > 0$, $T[0, k, m]$ and $T[1, k, m]$ have distinct output at the root for the following query:

$$Q(v) := \neg \left( \exists^{\geq 1} y \left( E(y, v) \wedge \left( \neg \left( \exists^{\geq 2} v \left( E(v, y) \wedge \top \right) \right) \right) \right) \right) = \forall y \left( E(y, v) \to \exists^{\geq 2} z E(z, y) \right)$$

expressing that the vertex $v$'s neighbors have degree at least 2. Note that GC2 can emulate other Boolean connectives (here $\to$) combining the ones it has, although it cannot use them directly. Clearly, $Q$ is a GC2 query of fixed depth 4. We can easily generalize to any GC2 query for which the root vertex of both trees have different output. In conclusion, the uniform expressivity of GNNs with superpolynomial bounded activation functions is weaker than those of the ReLUs.

## 5 Towards efficient non-uniform expressivity

Our second result focuses on *non-uniform* expressivity, where the size of the GNN is allowed to grow with the size of the input graphs. The result that we obtain holds for the wide class of activation functions defined below, and to any other activation function that can express them through a neural network. This class (See Lemma C.1 in Appendix C) is characterized by a fast convergence to the linear threshold activation function

$$\sigma_*(x) := \begin{cases} 0 & \text{if } x < 1/2 \\ 1/2 & \text{if } x = 1/2 \\ 1 & \text{if } x > 1/2 \end{cases} . \tag{2}$$

Recall that $\sigma'$ denotes the derivative of $\sigma$ and $\sigma^N = \sigma \circ \cdots \circ \sigma$ its $N$-th iterated composition.

**Definition 5.1** (Step-Like Activation Function). A *step-like activation function* is a $C^2$-map $\sigma : \mathbb{R} \to \mathbb{R}$ satisfying for $\eta \in [0, 1)$, $\varepsilon \in (0, 1/2)$, $N \in \mathbb{N}$ and $H > 0$ the following:

(a) $\sigma(0) = 0$ and $\sigma(1) = 1$.

(b) $|\sigma'(0)|, |\sigma'(1)| \leq \eta$.

(c) for all $x \notin (\varepsilon, 1 - \varepsilon)$, $|\sigma^N(x) - \sigma_*(x)| \leq \min\{\varepsilon, (1 - \eta)/H\}$.

(d) for all $x \in \sigma^N (\mathbb{R} \setminus (\varepsilon, 1 - \varepsilon))$, $|\sigma''(x)| \leq H$.

*Remark* 5.1. Seeing $\sigma : \mathbb{R} \to \mathbb{R}$ as a discrete dynamical system, (a) and (b) translate into 0 and 1 being attractive fixed points; (c) says that, after $N$ iterations, every point in $\mathbb{R} \setminus (-\varepsilon, \varepsilon)$, the complement of $(-\varepsilon, \varepsilon)$, is in the attractive region of either 0 or 1; and (d) provides a bound on a constant related to the size of these attractive regions.

Our main contribution presented Theorem 3.4 can be deduced from its technical version stated in the theorem below. All proof details are relegated to Appendix C.

**Theorem 5.1.** *Let $\sigma$ be a step-like activation function. Then, for every GC2 query $Q$ of depth $d$ and every positive integer $\Delta$, there is a GNN with activation function $\sigma$, $d$ iterations and size*

$$\left( 3 + N + \frac{2 + \log(\Delta + 2)}{1 - \log(1 + \eta)} \right) d$$

*that express $Q$ over all graphs with degree $\leq \Delta$. Moreover, if $\eta = 0$, the size of the GNN can be further reduced to*

$$(3 + N + \log(2 + \log(\Delta + 2))) d.$$

*Remark* 5.2. The logarithmic dependence of the GNN's size on the degree $\Delta$ of the considered class of graphs guarantees that the GNN has a reasonable size for applications. Moreover, if $\eta = 0$, then we have a log-log dependency, which for whichever practical setting is an almost constant function.

*Remark* 5.3. Our proof further shows that only the width of the combination function depends on the depth $d$ of the query, and that the number of layers remains independent of $d$.

Now, to give meaning to the above result, we will prove the following proposition giving us a way of constructing step-like activation functions out of $\tanh$ and Sigmoid.

**Proposition 5.2.** *There is a step-like activation function $\overline{\sigma}_{\mathrm{arctan}}$ with $\eta = 0.64$, $\varepsilon = 0.1$, $N = 0$ and $H = 1.52$ that can be expressed with a 3-layer NN with activation function $\arctan$ of size 3.*

**Proposition 5.3.** *There is a step-like activation function $\overline{\sigma}_{\tanh}$ with $\eta = 0$, $\varepsilon = 0.2$, $N = 0$ and $H = 2.2$ that can be expressed with a 4-layer NN with activation function $\tanh$ of size 6.*

**Proposition 5.4.** *There is a step-like activation function $\overline{\sigma}_{\mathrm{Sigmoid}}$ with $\eta = 0$, $\varepsilon = 0.2$, $N = 0$ and $H = 2.2$ that can be expressed with a 4-layer NN with activation function Sigmoid of size 6.*

Using these propositions, we get the following two important corollaries of Theorem 5.1 to get the almost-uniform expressivity of GC2 queries for $\tanh$ and Sigmoid, for which uniform expressivity is not possible due to Theorem 3.2.

**Corollary 5.5.** *For every GC2 query $Q$ of depth $d$ and every positive integer $\Delta$, there is a GNN with activation function $\arctan$, $d$ iterations and size at most $(10 + 3.5 \log(\Delta + 2)) d$ that expresses $Q$ over all graphs with degree $\leq \Delta$.*

**Corollary 5.6.** *For every GC2 query $Q$ of depth $d$ and every positive integer $\Delta$, there is a GNN with activation function $\tanh$, $d$ iterations and size at most $(9 + 4 \log(2 + \log(\Delta + 2))) d$ that expresses $Q$ over all graphs with degree $\leq \Delta$.*

**Corollary 5.7.** *For every GC2 query $Q$ of depth $d$ and every positive integer $\Delta$, there is a GNN with activation function Sigmoid, $d$ iterations and size at most $(9 + 4 \log(2 + \log(\Delta + 2))) d$ that expresses $Q$ over all graphs with degree $\leq \Delta$.*

## 6 NUMERICAL EXPERIMENTS

In this section, we demonstrate how the size of the neural networks impacts the GNN's ability to express GC2 queries via two experiments.

### 6.1 QUERIES

In our experiments, we consider the following two queries:

$$Q_1(v) := \neg \left(\exists^{\geq 1} y \left(E(y, v) \wedge \left(\neg \left(\exists^{\geq 2} v \left(E(v, y) \wedge \top\right)\right)\right)\right)\right) = \forall y \left(E(y, v) \to \exists^{\geq 2} z E(z, y)\right)$$

and

$$Q_2(v) = \mathsf{Red}(v) \wedge \left(\exists^{\geq 1} x \left(E(x, v) \wedge \left(\exists^{\geq 1} v \left(E(v, x) \wedge \mathsf{Blue}(v)\right)\right) \wedge \left(\exists^{\geq 1} v \left(E(v, x) \wedge \mathsf{Red}(v)\right)\right)\right)\right).$$

One can see that $Q_1(v)$ expresses that all neighbors of $v$ have degree at least two, and $Q_2(v)$ that $v$ is red and it has a neighbor that has a red neighbor and a blue neighbor. Moreover, $Q_1$ has depth 4 and $Q_2$ has depth 7. Note that $Q_1$ applies even to graphs with any number of colors, while $Q_2$ applies to graph having at least two colors.

### 6.2 EXPERIMENTAL SETUP

**Computing and approximating queries.** For the first experiment, our set of input graphs is composed of pairs of trees $(T[0, k, m], T[1, k, m])_{k \in \mathbb{N}, m \in \mathbb{N}}$ presented in Section 4. These are unicolored for $Q_1$ and colored as follows for $Q_2$: the root $s$ and the $x_i$ are red, and the leaves are blue. We compare for $i \in [2]$, the two following GNNs:

*(a)* The ReLU GNN that exactly computes the query $Q_i$.

We use the construction of (Barceló et al., 2020), specified in the proof of Theorem 5.1 where each of the GNNs has the same combination function in each iteration. For $Q_1$, the ReLU GNN has 4

iterations (depth of $Q_1$); and for $Q_2$, the ReLU GNN has 7 iterations (depth of $Q_2$). The combination functions are of the form

$$\mathsf{comb}_t(x, y) = \mathrm{ReLU}(Ax + By + C),$$

where $x, y \in \mathbb{R}^d$, $t \in \{0, \ldots, d - 1\}$ and $d = 4$ for $Q_1$ and $d = 7$ for $Q_2$. The parameters $A$, $B$ and $C$ are specified in Appendix D.

*(b)* The Pfaffian GNN with activation function: $\sigma : x \mapsto 1/2 + (2/\pi) \arctan(x - 1/2)$ that expresses $Q_i$ non uniformly.

We use the construction in the proof of Theorem 5.1, where as in the GNNs above, the combination functions are all the same. In this case, the combination functions are of the form

$$\mathsf{comb}_t(x, y) := \sigma^\ell(Ax + By + c)$$

where $x, y \in \mathbb{R}^d$, $t \in \{0, \ldots, d - 1\}$, $d = 4$ for $Q_1$ and $d = 7$ for $Q_2$, and the parameters $A$, $B$ and $c$ are the same as the ones of the ReLU counterparts, given in Appendix D. Note that this is a NN with $\ell + 2$ layers, where $\ell$ of the layers have width $d$ and one has width $2d$.

We report for different values of $\ell$ and number of vertices of the graphs inputs (which is function of $k$ and $m$), how close the two outputs of the Pfaffian GNNs, on the tree inputs $T[0, k, m]$ and $T[1, k, m]$. Since the root of these trees have respectively 0 and 1 as output of the query $Q$, the ReLU GNN constructed will have a constant gap of 1. Theorem 3.2 states that these outputs will tend to be arbitrarily close when the order of the graph increases, for a fixed $\ell$. In addition, Theorem 3.4 tells us that the number of iterations should impact exponentially well how large graphs can be handled to be at a given level of precision of a gap 1.

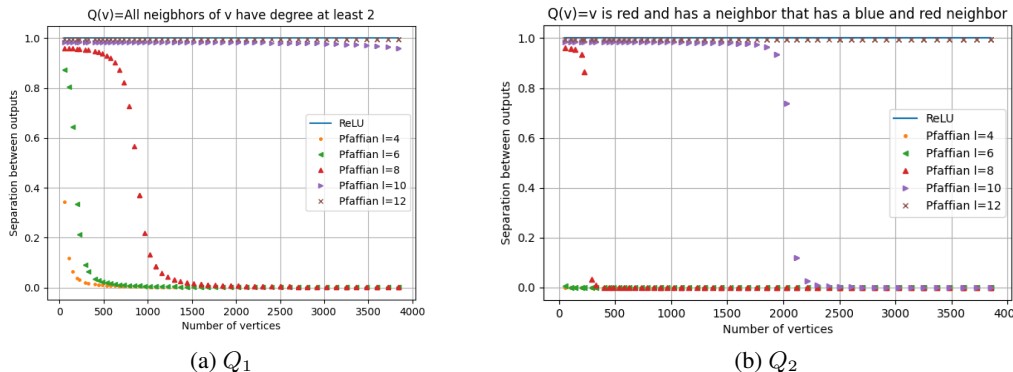

(a) $Q_1$           (b) $Q_2$

Figure 3: Separation power of $\mathrm{CReLU}$ GNN vs. Pfaffian GNN above as function of the graph size

**Learning queries.** For the second experiment, we compare the GNN's ability to learn GC2 queries, depending on the activation considered (ReLU vs. Pfaffian). We consider the same two queries as above and train two distinct GNNs: (a) A first GNN with $\mathrm{Sigmoid}$ activation function, and (b) A second GNN with $\mathrm{CReLU}$ activation functions. Both GNNs have 4 and 7 iterations when trained to learn the first and second query respectively. Each iteration is attributed his own combine function, a feedforward neural network with one hidden layer. This choice is justified by our theoretical results that one can either compute exactly a GC2 query (with the ReLU one), or approximate it efficiently (with the Pfaffian one) with only one hidden layer for the combine function. Our training dataset is composed of 3750 graphs of varying size (between 50 and 2000 nodes) of average degree 5, and generated randomly. Similarly, our testing dataset is composed of 1250 graphs with varying size between 50 and 2000 vertices, of average degree 5.

The experiments were conducted on a Mac-OS laptop with an Apple M1 Pro chip. The neural networks were implemented using PyTorch 2.3.0 and Pytorch geometric 2.2.0. The details of the implementation are provided in the supplementary material.

### 6.3 EMPIRICAL RESULTS

Figure 3 reports our results for the first experiment. As our theory predicts, the CReLU GNN designed with the appropriate architecture and weights is able to distinguish the source vertices of both trees with constant precision 1, regardless of the number of vertices. On the other end, the Pfaffian GNNs struggle to do separation for values of $\ell \leq 8$ (recall that $\ell$ is the number of compositions, i.e. the number of layers in the combinations functions) for graphs of than 1500 vertices. As the value of $\ell$ increases slightly to 10 for the first query and 12 for the second, the capacity of these GNNs dramatically increases to reach a plateau close to 1 for graphs up to 4000 vertices. This also illustrates our second theoretical result predicting that the step-like activation functions endow GNNs efficient approximation power of GC2 queries.

Figure 4 reports the mean square error on our test set for learning the same two queries. We observe that the error is similar for both GNNs, with a slight advantage for Pfaffian GNNs. This suggests that better uniform capacities does not imply that learning the queries is getting easier, given the same architecture and learning method. One potential reason is numerical: the combination functions using step-like activation functions used in our approximation of GC2 queries have likely to have large gradients. We suspect this adds to the difficulty to efficiently train these networks to optimality, despite their ability to approximate queries to an arbitrary level of precision.

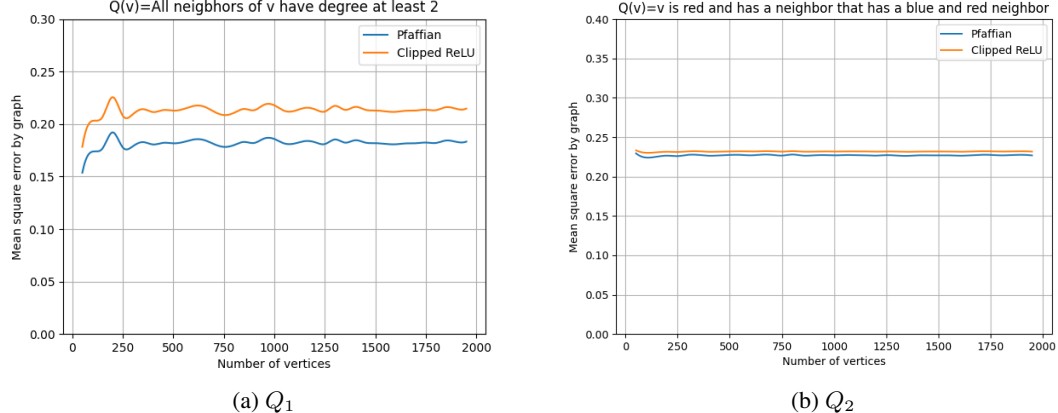

(a) $Q_1$          (b) $Q_2$

Figure 4: Learning queries with Pfaffian GNN vs ReLU GNN.
The mean square error per graph on the test set is displayed as a function of the order of the graph.

## 7 DISCUSSION AND OPEN QUESTIONS

Uniform expressivity is a desirable property when some information on the function or query to learn is available. Structural information may allow to design the GNN appropriately in a hope to express the query by selecting the correct weights, for example after learning from samples. In this study, we showed that many activation functions are not as powerful as CReLU GNNs from a uniform standpoint. This limitation is counterbalanced by the efficiency of many activations (including the Pfaffian ones that are weaker than CReLU) for non-uniform expressivity, where one allows the number of parameters to be non-constant with respect to the input graphs. Therefore, we argue that despite being desirable, uniform expressivity has to be complemented by at different measures for a thorough study on the expressivity of GNNs.

Moreover, it is frequent that one does not know in advance the structure of the query to be learned. Even when knowing the appropriate architecture for which approximate expressivity is guaranteed, learning the appropriate coefficients can be challenging as illustrated in our numerical experiments. However, these experiments are only preliminary and constitute by no means an exhaustive exploration of all the architectures and hyperparameters to learn those queries. As such, we hope this study will constitute a first step towards provably efficient and expressive GNNs, and suggest new directions to bridge the gap between learning and expressivity.

ACKNOWLEDGMENTS AND DISCLOSURE OF FUNDING

Sammy Khalife gratefully acknowledges support from Air Force Office of Scientific Research (AFOSR) grant FA95502010341, and from the Office of Naval Research (ONR) grant N00014-24-1-2645. Josué Tonelli-Cueto is thankful for the generous generous support of the Acheson J. Duncan Fund for the Advancement of Research in Statistics, grant 25-28. Josué Tonelli-Cueto thanks Jazz G. Suchen for useful discussions during the write-up of this paper, particularly regarding Proposition C.3, and Brittany Shannahan and Lewie-Napoleon III for emotional support.

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

## A  FACTS ABOUT PFAFFIAN FUNCTIONS

The following three results are folklore in the theory of Pfaffian functions. We finish with a proof of Proposition 2.1.

The first proposition is just a direct consequence of the so-called *o-minimality* of Pfaffian functions Wilkie (1999), which guarantees that Pfaffian systems share with polynomial systems their finiteness properties—among which we have the finiteness of zero-dimensional zero sets. The first lemma is an easy consequence of the chain rule and the definition of Pfaffian functions. The second and third lemmas follow from the proposition, but we include a proof for completeness.

**Proposition A.1.** *Let $f : \mathbb{R} \to \mathbb{R}$ be a Pfaffian function. Then $f$ is either constant or has finitely many zeroes in $\mathbb{R}$.* $\qquad\square$

**Lemma A.2.** *The addition, multiplication, composition and derivatives of Pfaffian functions are Pfaffian.* $\qquad\square$

**Lemma A.3.** *Let $f : \mathbb{R} \to \mathbb{R}$ be a Pfaffian and bounded function. Then $f$ has a finite limit in $+\infty$.*

*Proof.* By Lemma A.2, since $f$ is Pfaffian, so is $f'$. Hence, by Proposition A.1, either $f'$ is constant, and so $f' = 0$, or $f'$ has finitely many zeros. In the first case, $f$ is constant and so it trivially has a limit. In the second case, let $R > 0$ be such that $(R, \infty)$ does not contain any zero of $f'$. Then either $f'$ is strictly positive or strictly negative on $(R, \infty)$, and so either $f$ is strictly increasing or strictly decreasing on $(R, \infty)$. Hence, by the supremum axiom of the real numbers, we have that

$$\lim_{x \to \infty} f(x) = \sup\{f(y) \mid y \in (R, \infty)\}$$

if $f$ is strictly increasing on $(R, \infty)$, or

$$\lim_{x \to \infty} f(x) = \inf\{f(y) \mid y \in (R, \infty)\}$$

if $f$ is strictly decreasing on $(R, \infty)$. Since $f$ is bounded, the right-hand side is finite in both cases. Thus $f$ has a finite limit in $+\infty$. $\qquad\square$

**Lemma A.4.** *A bounded Pfaffian function $f : \mathbb{R} \to \mathbb{R}$ has a bounded derivative.*

*Proof.* By Lemma A.2, $f'$ and $f''$ are Pfaffian, because $f$ is so. Thus, by Proposition A.1, $f$, $f'$ and $f''$ have a finite number of zeros—if any of them are constant, then $f$ cannot be bounded. Hence there is $R > 0$ such that all the zeros of $f$, $f'$ and $f''$ are inside $(-R, R)$. In this way, since $f'$ is bounded on any bounded interval, we only need to prove that $f'$ is bounded on $[R, \infty)$ and on $(-\infty, -R]$. Without loss of generality, we will focus on the former case.

Assume, without loss of generality, after multiplying $f$ and the variable $X$ by $-1$ if necessary, that $f(R) > 0$ and $f'(R) > 0$. Now, we either have that $f''$ is positive or negative on the whole $[R, \infty)$,

and so for all $x \in [R, \infty)$, $f'(x) \geq f'(R) > 0$, if $f'$ positive, or $0 < f'(x) \leq f(R)$, if $f'$ negative. In the first case, for $x \geq R$,

$$f(x) = f(R) + \int_R^x f'(s)\,\mathrm{d}s \geq f(R) + (x - R)f'(R)$$

goes to infinity as $x \to \infty$, which contradicts our assumption on $f$. Thus, we are in the second case, in which $f'$ is bounded as desired. $\qquad\square$

Now, we end with a proof of Proposition 2.1.

*Proof of Proposition 2.1.* The only part requiring proof is that they are Pfaffian, that they are bounded is easily shown.

We have that $\tanh'(x) = 1 - \tanh(x)^2$ and that $\mathrm{Sigmoid}'(x) = -e^{-x}\,\mathrm{Sigmoid}(x)^2$. In this way,

$$\tanh'(x) = P(x, \tanh(x))$$

with $P(x, y) = 1 - y^2$, and

$$\mathrm{Sigmoid}'(x) = P(x, e^{-x}, \mathrm{Sigmoid}(x)), \ e^{-x} = Q(x, e^{-x})$$

with $P(x, y_1, y_2) = -y_1 y_2^2$ and $Q(x, y) = -y$, show that $\tanh$ and $\mathrm{Sigmoid}$ are Pfaffian.

Alternatively, since

$$\tanh(x) = 2\,\mathrm{Sigmoid}(2x) - 1 \ \text{ and } \ \mathrm{Sigmoid}(x) = \tanh(x/2)/2, \tag{3}$$

it would have been enough to show that either $\tanh$ or $\mathrm{Sigmoid}$ is Pfaffian to conclude that the other is so, by Lemma A.2. $\qquad\square$

## B  PROOF OF THEOREM 3.2 AND COROLLARY 3.3

*Proof of Theorem 3.2.* We consider the set of activation functions

$$\Xi := \{\sigma : \mathbb{R} \to \mathbb{R} : \sigma \text{ is Pfaffian, bounded and superpolynomial}\}$$

where the notions above where given in Definitions 2.5 and 3.2. Let $S_r$ be the set of $\mathbb{R}^r \to \mathbb{R}$ functions of fast decay in the last variable:

$$S_r := \{\phi : \mathbb{R}^{1+(r-1)} \to \mathbb{R} : \forall P \in \mathbb{R}[X], \ \forall x \in \mathbb{R} \ \lim_{y \to +\infty} |P(y_{r-1})\phi(x, y_1, \cdots, y_{r-1})| = 0\}$$

where for functions $F : \mathbb{R}^d \to \mathbb{R}$, $\lim_{z \to +\infty} F(z) = 0$ means that for every $\epsilon > 0$ there exists $\beta \geq 0$, such that $\min(z_1, \cdots, z_d) \geq \beta$ implies $|F(z)| < \epsilon$.

For non-negative integers $k$ and $m$, let $T[x, k, m]$ be the rooted tree of Figure 2 in which the root $s$ has $m$ descendants $x_i$, having $x_i$ $x$ descendants if $i = 1$ and $k$ descendants if $i \geq 2$. For ease of notation, we will refer to $T[x, k, m]$ simply as $T$ and to any of the descendants of $x_i$ as $l_i$. For any non-negative integer $t$ and $v \in V(T)$, let $\xi^t(v, T)$ be the embedding returned for node $v$ by a GNN with activation function $\sigma \in \Xi$ after $t$ iterations. Let $M > 0$ be an integer. We will prove by induction on $t$ that for any $t \in \{1, \cdots, M\}$:

$$\begin{aligned}
\xi^t(s, T) &= v_t + \eta_t(x, k, m) \\
\xi^t(x_{i \geq 2}, T) &= g_t(k) + \epsilon_t(x, k, m) \\
\xi^t(l_{i \geq 2}, T) &= h_t(k) + \nu_t(x, k, m) \\
\xi^t(x_1, T) &= g_t(k) + \epsilon_t^1(x, k, m) \\
\xi^t(l_1, T) &= h_t(k) + \nu_t^1(x, k, m)
\end{aligned}$$

with the following properties: a) $v_t \in \mathbb{R}^{d_t}$ is constant with respect to $x$ and $m$. b) each coordinate of $g_t$ and $h_t$ are bounded Pfaffian functions of $k$ that depend only on the combination and aggregation functions and the iteration $t$, and c) each coordinate of $\nu_t, \eta_t, \epsilon_t, \epsilon_t^1, \nu_t^1$ are in $S_3$.

In the remaining of the proof we explicitly treat the case of one-dimensional embeddings (i.e. $d_t = 1$ for every $t \in [T]$) for ease of notation. Our proof extends to multi-dimensional embeddings with the three additional ingredients:

i) considering only one input variable and the rest fixed, each coordinate of a combine function given by a neural network with an activation function $\sigma \in \Xi$ is a bounded Pfaffian function that is constant or superpolynomial (see Lemma A.2). Therefore, one can also use in that case Corollary A.1 on the zeroes of these functions, as we do in the one-dimensional case.

ii) the error terms of each signal can be controlled using norms of the Jacobian matrix instead of the gradient, as we shall see next.

iii) Given a bounded *multivariate* Pfaffian function $f$, for any fixed reals $\lambda_1, \cdots, \lambda_r$, $f(\lambda_1 m, \lambda_2 m, \cdots, \lambda_r m)$ has a limit when $m \to +\infty$. This follows from the fact that the univariate function $x \mapsto f(\lambda_1 x, \lambda_2 x, \cdots, \lambda_r x)$ is Pfaffian bounded.

***Base case:*** Obvious as all embeddings are initialized to 1.

***Induction hypothesis:*** Suppose that for some integer $t \geq 0$, for every $u \leq t$:

   i) The above system of equation is verified.

   ii) $g_u, h_u$ are Pfaffian and bounded functions.

   iii) $\nu_u, \eta_u, \epsilon_u, \epsilon_u^1$ and $\nu_u^1 \in S_3$.

***Induction step:*** For the remaining of the proof, for convenience we drop the dependency on $t$ of $\mathsf{comb}_t$ and simply write $\mathsf{comb}$, which is a neural network with some activation $\sigma \in \Xi$. First note that $\mathsf{comb}$ has bounded derivatives as the derivative of a bounded Pfaffian function is also bounded, see Lemma A.4. Let $K_{\mathsf{comb}} := \sup_{x \in \mathbb{R}^2} \|\partial_{x_1} \mathsf{comb}(x), \partial_{x_2} \mathsf{comb}(x)\|_2$.

The update rule for the leaves writes
$$\begin{aligned}
\xi^{t+1}(l_{i\geq 2}, T) &= \mathsf{comb}(\xi^t(l_i, T), \xi^t(x_i, T)) \\
&= \mathsf{comb}(h_t(k) + \nu_t(x, k, m), g_t(k) + \epsilon_t(x, k, m)) \\
&:= \mathsf{comb}(h_t(k), g_t(k)) + \nu_{t+1}(x, k, m) \\
&:= h_{t+1}(k) + \nu_{t+1}(x, k, m)
\end{aligned}$$
where
$$\begin{aligned}
|\nu_{t+1}(x, k, m)| &= |\mathsf{comb}(h_t(k) + \nu_t(x, k, m), g_t(k) + \epsilon_t(x, k, m)) - \mathsf{comb}(h_t(k), g_t(k))| \\
&\leq K_{\mathsf{comb}}(\nu_t(x, k, m), \epsilon_t(x, k, m))\|_2 \\
&\leq K_{\mathsf{comb}} \sqrt{2} \|(\nu_t(x, k, m), \epsilon_t(x, k, m))\|_\infty
\end{aligned}$$
similarly, $\xi^{t+1}(l_1, T) = \mathsf{comb}(h_t(k), g_t(k)) + \nu_{t+1}^1(x, k, m)$ where
$$\nu_{t+1}^1(x, k, m) = |\mathsf{comb}(h_t(k) + \nu_t^1(x, k, m), g_t(k) + \epsilon_t^1(x, k, m)) - \mathsf{comb}(h_t(k), g_t(k))|$$
For the intermediary vertices, we have
$$\begin{aligned}
\xi^{t+1}(x_{i\geq 2}, T) &= \mathsf{comb}(\xi^t(x_i, T), \xi^t(s, T) + k\xi^t(l_i, T)) \\
&= \mathsf{comb}(g_t(k) + \epsilon_t(x, k, m), v_t + \eta_t(x, k, m) + k(h_t(k) + \nu_t(x, k, m))) \\
&= \mathsf{comb}(g_t(k), v_t + kh_t(k)) + \epsilon_{t+1}(x, k, m) \\
&:= g_{t+1}(k) + \epsilon_{t+1}(x, k, m)
\end{aligned}$$
Let $\Delta := \eta_t(x, k, m) + k\nu_t(x, k, m)$
$$\begin{aligned}
|\epsilon_{t+1, m}(k)| &= |\mathsf{comb}(g_t(k) + \epsilon_t(x, k, m), v_t + \Delta + kh_t((k)) - \mathsf{comb}(g_t(k), v_t + kh_t(k))| \\
&\leq K_{\mathsf{comb}} \sqrt{2} \|(\epsilon_t(x, k, m), \Delta)\|_\infty
\end{aligned}$$
Similar inequalities can be derived for $\xi^t(x_1, T)$. Finally, for the root we have
$$\xi^{t+1}(s, T) = \mathsf{comb}(\xi^t(s, T), \sum_{i=1}^{\ell} \xi^t(x_i, T))$$
$$\begin{aligned}
&= \mathsf{comb}(v_t + \eta_t(x, k, m), g_t(x) + (m-1)g_t(k) + \epsilon_t^1(x, k, m) + (m-1)\epsilon_t(x, k, m)) \\
&:= \mathsf{comb}(v_t, g_t(x) + (m-1)g_t(k)) + \eta_{t+1}(x, k, m)
\end{aligned}$$

Using Corollary A.1, for every $u \in \{1, \cdots, t+1\}$, $g_u$ is either constant equal to $0$ on $\mathbb{R}$ or has no zeroes after some rank. Therefore, there exists an integer $k^\star$ such that for every $k_i \geq k^\star$ and every $u \in \{1, \cdots, t+1\}$, one of the following cases holds:

Case a) $g_u(k_i) > 0$.

Case b) $g_u(k_i) < 0$.

Case c) $g_u = \tilde{0}$ on $\mathbb{R}$.

We remind the reader that a similar case discussion here extends to the multidimensional case, i.e. when the function $g_u$ take vectorial values, and when the combine function has multi-dimensional co-domain, see for example comment iii). Note that we can select a common $\beta = k^\star$ for each iteration. Now, suppose that $|k| \geq \beta$.

By continuity of comb and $g_t$ having a limit in $+\infty$,

$$\forall \epsilon > 0, \ \exists M, \ \forall k \geq \beta, \ \forall m \geq M, \ |\mathsf{comb}(v_t, g_t(x) + (m-1)g_t(k)) - l| \leq \epsilon$$

where $l = \lim_{y \to +\infty} \mathsf{comb}(v_t, y)$ if case a) and $l = \lim_{y \to -\infty} \mathsf{comb}(v_t, y)$ if case b), and $l = \mathsf{comb}(v_t, 0)$ otherwise (case c). Hence

$$\xi^{t+1}(s, T) = \mathsf{comb}(v_t, g_t(x) + (m-1)g_t(k)) + \eta_{t+1}(x, k, m)$$

$$= \begin{cases} \lim_{y \to -\infty} \mathsf{comb}(v_t, y) + \eta_{t+1}(x, k, m) & \text{if } g_t(k^\star) < 0 \\ \lim_{y \to +\infty} \mathsf{comb}(v_t, y) + \eta_{t+1}(x, k, m) & \text{if } g_t(k^\star) > 0 \\ \mathsf{comb}(v_t, 0) + \eta_{t+1}(x, k, m) & \text{otherwise} \end{cases}$$

where $\eta_{t+1} = \eta_{t+1}^1 + \eta_{t+1}^2$. In the third case the induction step goes through immediately. In the following, we suppose without loss of generality that $g_t(k^\star) > 0$, with

$$|\eta_{t+1}^1(x, k, m)| := |\mathsf{comb}(v_t + \eta_t(x, k, m), g_t(x) + (m-1)g_t(k) + \epsilon_t^1(x, k, m)$$
$$+ (m-1)\epsilon_t(x, k, m)) - \mathsf{comb}(v_t, g_t(x) + (m-1)g_t(k))|$$

and

$$|\eta_{t+1}^{(2)}(x, k, m)| := \left| \mathsf{comb}(v_t, g_t(x) + (m-1)g_t(k)) - \lim_{y \to +\infty} \mathsf{comb}(v_t, y) \right|$$

First, it is clear that $\eta_{t+1}^2(x, k, m)$ can be made less than $\epsilon$ for some $m$, due to the existence of the limit $\lim_{y \to +\infty} \mathsf{comb}(v_t, y)$ and that $g_t(k)$ is chosen to be non zero (for every $k$ after rank $k^\star$ chosen above). Furthermore, $\eta_{t+1}^{(2)}$ is still in $S_3$ as for every fixed $x, k \geq k^\star$ the function $m \mapsto \mathsf{comb}(v_t, g_t(x) + (m-1)g_t(k))$ tends superpolynomially fast to its limit, due to the assumption made on the activation function and again from the fact that $g_t(k) \neq 0$ for every $k \geq k^\star$. Second,

$$|\eta_{t+1}^1(x, k, m)| \leq K_{\mathsf{comb}} \sqrt{2} \|(\eta_t(x, k, m), \epsilon_t^1(x, k, m) + (m-1)\epsilon_t(x, k, m))\|_\infty$$

We see that for this part the error terms $\epsilon_t^1$ and $\epsilon_t$ get multiplied by $m$. We know by the induction hypothesis that for every polynomial $P$,

$$\forall \epsilon > 0, \exists \beta, \ \forall m \geq \beta, \ \beta \leq |k| \implies |P(m)\epsilon_t(x, k, m)| \leq \epsilon.$$

In these conditions $|(m-1)\epsilon_t(x, k, m)| \leq \epsilon$. The property is still verified at rank $t+1$ as $|P(m)(m-1)\epsilon_t(x, k, m)| \leq |\underbrace{P(m)(m-1)}_{Q(m)}|\epsilon_t(x, k, m)$ where $Q(X) := (X-1)P(X)$ is a polynomial.

By the triangle inequality, $|\eta_{t+1,m}(k)| \leq |\eta_{t+1,m}^{(1)}(k)| + |\eta_{t+1,m}^{(2)}(k)|$, and so the desired property follows (after splitting the $\epsilon$ in half) and ends the induction. $\qquad \square$

*Remark* B.1. The above proof can be generalized for more general GNNs that use aggregation functions more general than sums. In particular, the proof above can be adapted to any aggregation function $\mathsf{agg}_t$ that admits a limit in $\{-\infty, +\infty\}$ when the size of the multiset with a repeated entry grows to $+\infty$.

*Proof of Corollary 3.3.* This follows from Theorem 3.2 and Proposition 2.1. $\qquad \square$

## C  PROOFS FOR SECTION 5

### C.1  PROOF OF THEOREM 5.1

Let us overview the proof of Theorem 5.1. First, we will show that the constructive proof of (Barceló et al., 2020, Proposition 4.2) for CReLU holds for the linear threshold activation function $\sigma_*$ defined in equation 2. After this, we will exploit that for a fast convergence of step-like function (see Definition 5.1) $\sigma$, $\sigma^\ell$ converges fast to $\sigma_*$ as the following lemma shows.

**Lemma C.1.** *Let $\sigma$ be a step-like activation function. Then for all $x \notin (\varepsilon, 1 - \varepsilon)$ and all $\ell \geq N$,*

$$\left| \sigma^\ell(x) - \sigma_*(x) \right| \leq \varepsilon \left( \frac{1 + \eta}{2} \right)^{\ell - N}. \tag{4}$$

*Moreover, if further $\eta = 0$, then for all $x \notin (\varepsilon, 1 - \varepsilon)$ and all $\ell \geq N$,*

$$\left| \sigma^\ell(x) - \sigma_*(x) \right| \leq \frac{\varepsilon}{2^{2^{\ell - N} - 1}}. \tag{5}$$

*Proof of Theorem 5.1.* Let $Q$ be a query of GC2 of depth $d$, and let $(Q_1, \ldots, Q_d)$ be an enumeration of its subformulas, being each $Q_i$ an atomic formula or a formula build from the previous ones. Without loss of generality, assume that $Q$ uses all $\ell \leq d$ colors and that the first $\ell$ subformulas $Q_1, \ldots, Q_\ell$ are the atomic formulas

$$\mathsf{Col}_1(x), \ldots, \mathsf{Col}_\ell(x)$$

corresponding to these $\ell$ colors. If this is not the case, then $Q$ involves $\ell' \leq \ell$ colors, say $\lambda_1, \ldots, \lambda_{\ell'}$, and so in the sequence $(Q_1, \ldots, Q_d)$, we will only have the atomic formulas $\mathsf{Col}_{\lambda_1}(x), \ldots, \mathsf{Col}_{\lambda_{\ell'}}(x)$ appearing. Thus to make the construction below work, we only need to reorder the subformulas and to perform the constructions below changing the color $\lambda_i$ to the color $i$ in the formulas and change the $A$ and $B$ in the first combination function $\mathsf{comb}_0$ from $A$ and $B$ to $AP$ and $BP$ where $P$ is the coordinate projection mapping $e_{\lambda_i}$ to $e_i$.

First, we extend the constructive proof of (Barceló et al., 2020, Proposition 4.2) to GNNs with linear threshold activations $\sigma_*$. In this extension, we have to make sure we can control the error made efficiently[3]. The GNN that we will build has $d$ iterations, being each combination functions of the form

$$\mathsf{comb}_t : (x, y) \mapsto \sigma_*(Ax + By + C) \qquad (t \in \{0, \ldots, d - 1\})$$

where $A$ and $B$ are $d \times d$ matrices and $C$ a $d$-dimensional vector $C$. The $j$th row of $A$, $B$ and $C$ will correspond to the corresponding subformula $Q_j$ and they will be filled as follows:

1) If $Q_j$ is the atomic formula corresponding to the color $\lambda_j$, then $A_{j,\lambda_j} = 1$ for $k = \lambda_j$ and $A_{j,k} = 0$ otherwise, $B_{j,k} = 0$ for all $k \in [d]$, and $C_j = 0$. Note that by our assumption, the first $\ell$ rows of $A$ are those of the identity matrix, and so the first $\ell$ components of the embeddings give the color of the vertex.

2) If $Q_j(x) = \exists^{\geq K} y\, (E(x, y) \wedge Q_i(y))$ for some $i < j$, then $A_{j,k} = 0$ for all $k \in [d]$, $B_{j,i} = 1$ and $B_{j,k} = 0$ for $k \neq i$, and $C_j = -(K - 1)$.

3) If $Q_j(x) = \exists^{\geq K} y\, (E(x, y) \wedge \mathsf{T})$, then $A_{j,k} = 0$ for all $k \in [d]$, $B_{j,k} = 1$ for $k \in [\ell]$ and $B_{j,k} = 0$ for $k \notin [\ell]$, and $C_j = -(K - 1)$.

4) If $Q_j(x) = \neg Q_i(x)$ with $i < j$, then $A_{j,i} = -1$ and $A_{j,k} = 0$ for $k \neq i$, $B_{j,k} = 0$ for all $k \in [d]$ and $C_j = 1$.

5) If $Q_j(x) = \neg \mathsf{T}$, then $A_{j,i} = B_{j,i} = 0$ for all $k \in [d]$ and $C_j = 0$.

6) If $Q_j(x) = Q_{i_1}(x) \wedge Q_{i_2}(x)$ for some $i_1, i_2 < j$, then $A_{j,i_1} = A_{j,i_2} = 1$ and $A_{j,k} = 0$ for $k \neq i_1, i_2$, $B_{j,k} = 0$ for all $j \in [d]$ and $C_j = -1$.

7) If $Q_j(x) = Q_i(x) \wedge \mathsf{T}$ or $Q_j(x) = \mathsf{T} \wedge Q_i(x)$ for some $i < j$, then $A_{j,i} = 1$ and $A_{j,k} = 0$ for $k \neq i$, $B_{j,k} = 0$ for all $j \in [d]$ and $C_j = 0$.

---

[3]The original proof to exactly compute the query can actually be extended for any function $g : \mathbb{R} \to \mathbb{R}$ verifying: there exists real $a < b$ such that i) for every real $x \leq a$, $g(x) = 0$, ii) for every real $x \geq b$, $g(x) = 1$ and iii) $g$ is non-decreasing on $[a, b]$.

8) If $Q_j(x) = \text{T} \land \text{T}$, then $A_{j,k} = B_{j,k} = 0$ for all $j \in [d]$ and $C_j = 1$.

For any integer $t \in \{0, \ldots, d-1\}$ and for every graph $G$ and each vertex $v \in V(G)$ of $G$, consider

$$q^{t+1}(v, G) := \text{comb}_t\left(q^t(v, G), \sum_{w \in \mathcal{N}_G(v)} q^t(w, G)\right). \tag{6}$$

Observe that for each $t \in [d]$, $q^t(v, G) \in \{0, 1\}^d$. With the exact same inductive argument as in (Barceló et al., 2020, Proposition 4.2), we have that for all $t \in [d]$ and all $j \in [t]$,

$$q_j^t(v, G) = Q_j(v, G),$$

i.e., the first $t$ components of the embedding $q^t(v, G)$ are equal to the first $t$ subformulas $Q_1, \ldots, Q_t$ of $Q$.

Now, we will now consider the GNN where we have substituted $\sigma_*$ by its approximation $\sigma^\ell$. The new GNN will have combination functions of the form

$$\text{comb}_t^\ell : (x, y) \mapsto \sigma^\ell(Ax + By + C) \qquad (t \in \{0, \ldots, d-1\})$$

with $A$, $B$ and $C$ as above, and it will construct the vertex embedding $\xi_t^\ell(v, G)$ given by the update rule

$$\xi_{t+1}^\ell(v, G) = \text{comb}_t^\ell\left(\xi_t^\ell(v, G), \sum_{w \in \mathcal{N}_G(v)} \xi_t^\ell(w, G)\right).$$

Observe that $\text{comb}_t^\ell$ is a neural network with input size $2d$, output size $d$, $\ell + 1$ layers and widths $2d, d, \ldots, d$, and so size $(\ell + 2)d$.

Let $\Delta$ be a positive integer and fix a graph $G$ with maximal degree bounded by $\Delta$. For $t \in \{0, \ldots, d\}$, we define

$$\epsilon_t := \max_{v \in V(G)} \|\xi^t(v, G) - q^t(v, G)\|_\infty.$$

We shall prove that, under the assumptions of the theorem, the following claim:

*Claim.* For any $t \in \{0, \ldots, d-1\}$,

$$\text{if } 2(\Delta + 2)\epsilon_t < 1, \text{ then } \epsilon_{t+1} \leq \begin{cases} \left(\frac{1+\eta}{2}\right)^{\ell-N} \varepsilon & \text{if } \eta > 0 \\ \frac{\varepsilon}{2^{2^{\ell-N}-1}} & \text{if } \eta = 0 \end{cases}$$

Under this claim, using induction, we can easily show that if either

$$\eta > 0 \text{ and } \ell \geq N + \frac{2 + \log(\Delta + 2)}{1 - \log(1 + \eta)}$$

or

$$\eta = 0 \text{ and } \ell \geq N + \log(2 + \log(\Delta + 2)),$$

then $\epsilon_d \leq 1/3 < 1/2$. Effectively, under any of the above assumptions, we have that

$$2(\Delta + 2)\epsilon_t < 1 \implies 2(\Delta + 2)\epsilon_{t+1} < 1$$

by the claim and the fact that $\varepsilon < 1/2$. Since $\epsilon_0 = 0$, we conclude, by induction, that

$$\epsilon_d \leq \begin{cases} \left(\frac{1+\eta}{2}\right)^{\ell-N} \varepsilon & \text{if } \eta > 0 \\ \frac{\varepsilon}{2^{2^{\ell-N}-1}} & \text{if } \eta = 0 \end{cases} \leq 1/3 < 1/2,$$

providing the expressivity. Note that the size of the above GNN is $2d + m$, and so the statement of the theorem follows.

Now, we prove the claim. For $t \in \{0, \ldots, d-1\}$ we have, by the triangle inequality, that

$$\epsilon_{t+1} = \max_{v \in V(G)} \|\xi^{t+1}(v, G) - q^{t+1}(v, G)\|_\infty$$

$$= \max_{v \in V(G)} \left\| \mathsf{comb}_t^\ell \left( \xi^t(v, G), \sum_{w \in \mathcal{N}_G(v)} \xi^t(w, G) \right) - \mathsf{comb}_t \left( q^t(v, G), \sum_{w \in \mathcal{N}_G(v)} q^t(w, G) \right) \right\|_\infty$$

$$\leq \max_{v \in V(G)} \left\| \mathsf{comb}_t^\ell \left( \xi^t(v, G), \sum_{w \in \mathcal{N}_G(v)} \xi^t(w, G) \right) - \mathsf{comb}_t \left( \xi^t(v, G), \sum_{w \in \mathcal{N}_G(v)} \xi^t(w, G) \right) \right\|_\infty$$

$$+ \max_{v \in V(G)} \left\| \mathsf{comb}_t \left( \xi^t(v, G), \sum_{w \in \mathcal{N}_G(v)} \xi^t(w, G) \right) - \mathsf{comb}_t \left( q^t(v, G), \sum_{w \in \mathcal{N}_G(v)} q^t(w, G) \right) \right\|_\infty.$$

The first maximum of the last inequality is bounded above by

$$\sup \left\{ \left\| \mathsf{comb}_t^\ell(x, y) - \mathsf{comb}_t(x, y) \right\|_\infty : \text{ for all } i, \ x_i \notin (\epsilon_t, 1 - \epsilon_t), \ y_i \notin (\Delta\epsilon_t, 1 - \Delta\epsilon_t) \right\},$$

since each component of $\xi^t(v, G)$ is at distance at most $\epsilon_t$ from $0$ or $1$ by definition of $\epsilon_t$. Now, $\mathsf{comb}_t^\ell(x, y) = \sigma^\ell(Ax + Bx + C)$ and $\mathsf{comb}_t(x, y) = \sigma_*(Ax + By + C)$ with $A, B \in \{-1, 0, 1\}^{d \times d}$ and $C \in \mathbb{Z}^d$. Moreover, the matrix $A$ has at most two non-zero entries per row, and $B$ at most one non-zero entry. Since the 1-norm of each row of $A$ is at most 2, and the one of $B$ is at most 1, we can improve the upper-bound with

$$\sup \left\{ |\sigma^\ell(z) - \sigma_*(z)| : z \notin ((\Delta + 2)\epsilon_t, 1 - (\Delta + 2)\epsilon_t) \right\},$$

By assumption, this quantity is bounded from above by the claimed by Lemma C.1.

For the second maximum, define

$$x := \xi^t(v, G) \text{ and } y := \sum_{w \in \mathcal{N}_G(v)} \xi^t(w, G)$$

and

$$x' := q^t(v, G) \text{ and } y' := \sum_{w \in \mathcal{N}_G(v)} q^t(w, G)$$

to simplify notation. By definition of $\epsilon_t$, we have

$$\|x - x'\|_\infty \leq \epsilon_t \text{ and } \|y - y'\|_\infty \leq \Delta\epsilon_t.$$

Now, arguing as above regarding $A_t$ and $B_t$, we have that

$$\|(Ax + By + C) - (Ax' + By' + C)\|_\infty = \|A(x - x') + B(y - y')\|_\infty \leq (2 + \Delta)\epsilon_t.$$

In this way, $Ax + By + C$ is a real vector with $\infty$-norm $< 1/2$ to the integer vector $Ax' + By' + C$ and so we conclude that

$$\mathsf{comb}(x, y) = \mathsf{comb}(x', y'),$$

because $\sigma_*$ takes the same value on an integer and on a real number at distance less than $1/2$ from it. Hence the value of the difference under the second maximum is zero, proving the claim. $\qquad \square$

To prove Lemma C.1 we will need the following well-known lemma.

**Lemma C.2.** *Let $f : \mathbb{R} \to \mathbb{R}$ be a $C^2$-function, $x_0 \in \mathbb{R}$ a fixed point of $f$ (i.e., $f(x_0) = x_0$) and $r > 0$. If $|f'(x_0)| = \eta < 1$, $\sup_{t \in [x_0 - r, x_0 + r]} |f''(t)| = H > 0$ and*

$$Hr \leq 1 - \eta,$$

*then for every integer $n \geq 0$ and $x \in [x_0 - r, x_0 + r]$,*

$$|f^n(x) - x_0| \leq \left( \frac{1 + \eta}{2} \right)^n |x - x_0| \leq \left( \frac{1 + \eta}{2} \right)^n r.$$

*Moreover, if $\eta = 0$, then for all $n$ and $x \in [x_0 - r, x_0 + r]$,*

$$|f^n(x) - x_0| \leq \left( \frac{H}{2} \right)^{2^n - 1} |x - x_0|^{2^n} \leq \frac{r}{2^{2^n - 1}}.$$

*Proof of Lemma C.1.* Fix $x \notin (\varepsilon, 1 - \varepsilon)$ and let $r = \min\{\varepsilon, (1 - \eta)/H\}$. By (c), we have that $|\sigma^N(x) - \sigma_*(x)| \leq r$ with $Hr \leq 1 - \eta$. Hence, since $\sigma_*(x)$ is a fixed point of $\sigma$, by (a), we have, by Lemma C.2, (b) and (e), that

$$|\sigma^\ell(\sigma^N(x)) - \sigma_*(x)|$$

converges to zero with the desired speed. □

*Proof of Lemma C.2.* Let $H = \sup_{t \in [x_0 - r, x_0 + r]} |f''(t)|$. Then we have that $r \leq \frac{1-\eta}{H}$ and so for $x \in [x_0 - r, x_0 + r]$, we will have, by Taylor's theorem, for some $s$ between $x$ and $x_0$,

$$f(x) = x_0 \pm \eta(x - x_0) + \frac{1}{2} f''(s)(x - x_0)^2,$$

where we write $x_0$ since $x_0 = f(x_0)$. Rearranging the expression and taking absolute values,

$$|f(x) - x_0| \leq \eta|x - x_0| + \frac{1}{2}H|x - x_0|^2 \leq \left(\eta + \frac{Hr}{2}\right)|x - x_0|,$$

since $|f''(s)| \leq H$ and $|x - x_0| \leq r$. Now, by assumption, $Hr/2 \leq \frac{1-\eta}{2}$, and so

$$|f(x) - x_0| \leq \left(\frac{1 + \eta}{2}\right)|x - x_0|.$$

For the case $\eta = 0$, we have instead

$$|f(x) - x_0| \leq \frac{H}{2}|x - x_0|^2.$$

Hence in both cases, induction on $n$ ends the proof. □

## C.2 Proofs of Propositions 5.2, 5.3 and 5.4

We now focus on Proposition 5.2, 5.3 and 5.4. First, we prove Proposition 5.2. Second, we show that Proposition 5.4 follows from Proposition 5.3. Third and last, we prove Proposition 5.3 by proving the more precise Proposition C.3.

*Proof of Proposition 5.2.* We just need to consider $\sigma_{\mathrm{arctan}}(x) = \frac{1}{2} + \frac{2}{\pi} \arctan(2x - 1)$. Then a simple computation shows that the given claim is true. □

*Proof of Proposition 5.4.* By equation 3, we can express $\tanh$ as a NN with activation function Sigmoid. Hence, we can transfor $\overline{\sigma}_{\mathrm{tanh}}$ into $\sigma_{\mathrm{Sigmoid}}$ by substituting each $x \mapsto \tanh(x)$ by $x \mapsto 2\,\mathrm{Sigmoid}(2x) - 1$. A Simple counting finishes the proof. □

Observe that Proposition 5.3 follows from the following precise proposition that we will prove instead.

**Proposition C.3.** *Let*

$$\sigma_{\mathrm{tanh}}(x) := \frac{1}{2} + \frac{\tanh(x - 1/2)}{2\tanh(1/2)}$$

*and*

$$\overline{\sigma}_{\mathrm{tanh}}(x) := \sigma_{\mathrm{tanh}}\left(\frac{\tanh(2ax - a) - \alpha\tanh(4ax - 2a)) + \tanh(6ax - 3a)}{2(\tanh(a) - \alpha\tanh(2a) + \tanh(3a))} + \frac{1}{2}\right)$$

*where $a \in (0.45, 0.46)$ and $\alpha \in (3.14, 3.15)$ are such that*

$$\min\left\{\frac{\mathrm{sech}^2(x) + 3\,\mathrm{sech}^2(3x)}{\mathrm{sech}^2(2x)} \mid x \in \mathbb{R}\right\} = \frac{\mathrm{sech}^2(a) + 3\,\mathrm{sech}^2(3a)}{\mathrm{sech}^2(2a)} = \alpha.$$

*Then $\sigma_{\mathrm{tanh}}$ and $\overline{\sigma}_{\mathrm{tanh}}$ are step-like activation functions with, respectively, $\eta = 0.86$, $\varepsilon = 0.16$, $N = 0$ and $H = 0.84$, and $\eta = 0$, $\varepsilon = 0.2$, $N = 0$ and $H = 2.2$.*

*Proof.* The claim about $\sigma_{\tanh}(x)$ follows after a simple numerical computation. For the claim about $\overline{\sigma}_{\tanh}(x)$ observe that

$$\overline{\sigma}_{\tanh}(x) = \sigma_{\tanh}(\tau(x))$$

where, by the choices above, we have that:

(0) $\tau(0) = 0$, $\tau(1) = 1$.

(1) $\tau$ is strictly increasing.

(2) $\tau'(x) = 0$ if and only if $x \in \{0, 1\}$.

Hence $\overline{\sigma}_{\tanh}$ is fixes 0 and 1, is strictly increasing and, by the chain rule, satisfies $\eta = 0$. The rest of the constants are found by numerical computation. $\square$

### C.3 PROOFS OF COROLLARIES 5.5, 5.6 AND 5.7 AND COMPOSITE NN

*Proof of Corollary 5.5.* This is a combination of Theorem 5.1 with Proposition 5.2 using Proposition C.4 below. $\square$

*Proof of Corollary 5.6.* This is a combination of Theorem 5.1 with Proposition 5.3 using Proposition C.4 below. $\square$

*Proof of Corollary 5.7.* This is a combination of Theorem 5.1 with Proposition 5.4 below. $\square$

**Proposition C.4.** *Let $\sigma : \mathbb{R} \to \mathbb{R}$ and $\tilde{\sigma} : \mathbb{R} \to \mathbb{R}$ be functions. If $\tilde{\sigma}$ can be written in terms of a NN with activation function $\sigma$ with $\ell + 2$ layers of widths $1, r_1, \ldots, r_\ell, 1$, then every NN with activation function $\tilde{\sigma}$ with $k + 1$ layers of widths $w_0, \ldots, w_{k+1}$ can be transformed into a NN with activation function $\sigma$ with $2 + \ell(k - 1)$ layers of widths*

$$w_0, w_1 r_1, \ldots, w_1 r_\ell, w_2 r_1, \ldots, w_2 r_\ell, \ldots, w_k r_1, \ldots, w_k r_\ell, w_{k+1}.$$

*In particular, if $\tilde{\sigma}$ can be written in terms of a NN with activation function $\sigma$ of size $r$, then every NN with activation function $\tilde{\sigma}$ of size $s$ can be transformed into a NN with activation function $\sigma$ of size at most $(r - 2)s$.*

*Proof.* Let the NN with activation function $\sigma$ have affine transformations $T_i : \mathbb{R}^{w_i} \to \mathbb{R}^{w_{i+1}}$ ($i \in \{0, \ldots, k\}$) and the NN with activation function $\sigma$ that expresses $\tilde{\sigma}$ have affine transformations $S_i : \mathbb{R}^{r_i} \to \mathbb{R}^{r_{i+1}}$ ($i \in \{0, \ldots, \ell + 1\}$), where $r_0 = r_{\ell+2} = 1$. Then, as an $\mathbb{R} \to \mathbb{R}$ map we have

$$\tilde{\sigma} = S_{\ell+1} \circ \sigma \circ S_\ell \circ \cdots S_1 \circ \sigma \circ S_0,$$

but, as a pointwise $\mathbb{R}^w \to \mathbb{R}^w$ map, we can write $\tilde{\sigma}$ as

$$\tilde{\sigma} = S_{\ell+1}^{\otimes w} \circ \sigma \circ S_\ell^{\otimes w} \circ \cdots S_1 \circ \sigma \circ S_0^{\otimes w}$$

where $S_i^{\otimes w}$ is the map $\mathbb{R}^{r_i w} \to \mathbb{R}^{r_{i+1} w}$ given by

$$\mathbb{R}^{r_i w} = (\mathbb{R}^{r_i})^w \ni \begin{pmatrix} x_1 \\ \vdots \\ x_w \end{pmatrix} \mapsto \begin{pmatrix} S_i(x_1) \\ \vdots \\ S_i(x_w) \end{pmatrix} \in (\mathbb{R}^{r_{i+1}})^w = \mathbb{R}^{r_{i+1} w}.$$

In this way, considering the NN with activation function $\sigma$ with affine transformation

$$S_0^{\otimes w_1} \circ T_0, S_1^{\otimes w_1}, \ldots, S_\ell^{\otimes w_1}, S_0^{\otimes w_2} \circ T_1 \circ S_{\ell+1}^{\otimes w_1}, S_1^{\otimes w_2}, \ldots, S_\ell^{\otimes w_2}, S_0^{\otimes w_3} \circ T_2 \circ S_{\ell+1}^{\otimes w_2},$$

$$\ldots,$$

$$S_0^{\otimes w_k} \circ T_{k-1} \circ S_{\ell+1}^{\otimes w_{k-1}}, S_1^{\otimes w_k}, \ldots, S_\ell^{\otimes w_k}, T_k \circ S_{\ell+1}^{\otimes w_k},$$

we transform the NN with activation function $\tilde{\sigma}$ into a NN with activation function $\sigma$ with the desired characteristics. The claim about the size is immediate. Note that if $r = 2$, then $\tilde{\sigma}$ is an affine map and the claim is still true. $\square$

# D  NUMERICAL EXPERIMENTS

**Computing and approximating queries.** For $Q_1$, the ReLU GNN has 4 iterations with $A$, $B$ and $c$ given by

$$A = \begin{pmatrix} 0 & 0 & 0 & 0 \\ -1 & 0 & 0 & 0 \\ 0 & 0 & 0 & 0 \\ 0 & 0 & -1 & 0 \end{pmatrix}, B = \begin{pmatrix} 1 & 0 & 0 & 0 \\ 0 & 0 & 0 & 0 \\ 0 & 1 & 0 & 0 \\ 0 & 0 & 0 & 0 \end{pmatrix}, C = \begin{pmatrix} -1 \\ 1 \\ 0 \\ 1 \end{pmatrix}.$$

Note that since we are in the unicoloring setting, all vectors are initialized to $e_1$.

For $Q_2$, the ReLU GNN has 7 iterations and $A$, $B$ and $c$ are given as:

$$A = \begin{pmatrix} 1 & 0 & 0 & 0 & 0 & 0 \\ 0 & 1 & 0 & 0 & 0 & 0 \\ 0 & 0 & 0 & 0 & 0 & 0 \\ 0 & 0 & 0 & 0 & 0 & 0 \\ 0 & 0 & 1 & 1 & 0 & 0 \\ 0 & 0 & 0 & 0 & 0 & 0 \\ 1 & 0 & 0 & 0 & 1 & 0 \end{pmatrix}, B = \begin{pmatrix} 0 & 0 & 0 & 0 & 0 & 0 \\ 0 & 0 & 0 & 0 & 0 & 0 \\ 1 & 0 & 0 & 0 & 0 & 0 \\ 0 & 1 & 0 & 0 & 0 & 0 \\ 0 & 0 & 0 & 0 & 0 & 0 \\ 0 & 0 & 0 & 1 & 0 & 0 \\ 0 & 0 & 0 & 0 & 0 & 0 \end{pmatrix}, C = \begin{pmatrix} 0 \\ 0 \\ 0 \\ 0 \\ -1 \\ 0 \\ -1 \end{pmatrix}.$$

Note that vertices are initialized to $e_1$ or $e_2$ depending on whether they are colored red or blue.

