# OpenReview forum: "Is uniform expressivity too restrictive? Towards efficient expressivity of GNNs"
_ICLR.cc/2025/Conference — ICLR 2025 Poster_

### Official Review · Reviewer_6aa7 · 2024-11-02

**Soundness:** 4
**Presentation:** 3
**Contribution:** 3
**Rating:** 8
**Confidence:** 3

**Summary:**

This papers studies the following fundamental question. Any node binary classification task can be defined as a logic formula. The paper tries to understand whether common GNN architectures can learn a given logic formulas **uniformly** (over all graphs with unbounded size). This is in stark contrast with prior research, which typically allows the model size to grow with input graph size. The authors gave both negative and positive results, showing that GNNs with sigmoid/tanh activations cannot uniformly approximate GC2 logic formulas, but they can almost-uniformly approximate them. The authors gave both counterexamples and constructions in their proof. Finally, they conduct experiments to verify their results.

**Strengths:**

I would like to say this is a quite interesting paper and I like it a lot. First, this paper studies GNN expressivity from a novel perspective. Unlike prior work which largely focuses on graph isomorphism (which is mainly TCS/graph theory perspectives), this paper is clearly more relevant to real neural network architectures instead of abstract color refinement algorithms, making it especially relevant to the community. On the other hand, this paper differs from traditional function approximate theory, as these theories often let the model size to grow in order to approximate better with a larger input domain. Instead, this paper gives both positive and negative results on whether a *fixed* model can deal with graph learning tasks for all inputs. This ability is unique for graph learning tasks.

Secondly, the paper provides a rigorous theoretical foundation. All definitions and statements are well-defined and accessible, which enhances clarity. I find the results nontrivial and impactful, and it enhances our understanding in the community.

**Weaknesses:**

One potential weakness of these results is that almost-uniform expressivity is perhaps already sufficient. Therefore, the limitations of sigmoid/tanh networks is not that severe in practice. So this paper may have a limited impact in guiding practical design of GNNs.

Update: I should acknowledge that I am not very familiar with first-order logics beyond the paper of Barcelo (2020). I am not sure if similar ideas have been raised in prior work (possibly under a different community).

**Questions:**

I am curious about the insights behind the theoretical results. I suspect this may be due to the sigmoid/tanh function is bounded, so it can not  express a quantity that grows with input size. On the other hand, ReLU/CReLU is not bounded. I wonder whether any unbounded non-linear activation is sufficient for uniform expressivity?


Miscellaneous：
1. Definition 2.4: U should be $U$.
2. Page 7, line 350: the word "the" is redundant.
3. What is the difference between Corollaries 5.6 and 5.7? I suspect there is a typo here.

---

> ### Author Response · Authors · 2024-11-22
>
> Thanks for the positive appreciation of our paper and your feedback!
>
> Regarding your comments about the weaknesses of the paper:
>
> a) Despite the uniform expressivity limitations, these same GNNs have indeed efficient approximation properties. Our results indeed indicate that uniform expressivity is only one measure that can be completed with efficient approximate expressivity. An additional insight is that, if some knowledge about the depth of GC2 query to be learned, then one can theoretically restrict to GNNs with combination functions of some depth. However, we agree that a fine-grained guidance on the design of GNNs would complement this work further.
>
> b) For the update part, to the best of our knowledge, such results do not exist in the exact same setup. We do an extended comparison of the existing results in the literature in the introduction that included the main results we are aware of at the interface of logic and expressivity of GNNs.
>
> Regarding your last question:
>
> Thanks for the interesting question. We are unable to provide a complete answer. However, let us point two observations: 1) Note that CReLU is bounded, and so boundedness alone is not an obstruction to uniform expressivity of GC2 formulas. 2) Also, as the paper "`The logic of rational graph neural networks", cited in the references, shows that unboundedness alone is not sufficient for uniform expressivity, since GNNs with rational activation functions cannot express uniformly GC2 formulas.
>
> Miscellaneous:
>
> 1) Thank you. We corrected it.
>
> 2) Thank you. We corrected it.
>
> 3) There is a typo in Corollaries 5.6 and 5.7, we have corrected them.

---

> > ### Comment · Reviewer_6aa7 · 2024-11-25
> > **Thank you**
> >
> > Thank you for your response. That's quite interesting to see that boundedness is not directly related to uniform expressivity. I am satisfied with the response and am happy to see the paper accepted.

---

### Official Review · Reviewer_Mu1a · 2024-11-03

**Soundness:** 2
**Presentation:** 3
**Contribution:** 3
**Rating:** 6
**Confidence:** 3

**Summary:**

In the study of GNN expressivity, it is crucial for practical usage to take the number of parameters needed into consideration. To this end, uniform expressivity has been proposed to test whether GNNs can express a query with the number of parameters independent on the size of the input graphs. However, in this paper the authors found that uniform expressivity tends to be too restricted for a wide class of GNN models. This paper then relaxes the concept of uniform expressivity into almost-uniform expressivity and further show that most GNNs can be still quite expressive when restricting the number of parameters to be logarithmic on the maximal degree, satisfying most practical needs. The results in this paper are justified by both theoretical analysis and numerical experiments.

**Strengths:**

- Compared with general expressivity measurement of GNNs, uniform expressivity sometimes tends to be too strict, thus provides negative results for GNNs: as shown in this paper, MPNNs with sigmoid activations cannot uniform express GC2 queries. To this end, it is nice to find a balance between general expressivity (which does not consider the number of parameters) and uniform expressivity (where the number of parameters is independ w.r.t. input graphs) for practical needs. The log complexity of parameters described in this paper is realizable for most practical situations.
- This paper considers a wide class of activations, defined as step-like activation functions. The authors show that their results are applicable for many practical activation functions including sigmoid and tanh.

**Weaknesses:**

- The study in this paper relies on the definition of step-like activation functions. The properties of step-like activation function still requires more discussion. Specially, I have some concerns regarding the definition of step-like activation functions:
  - The Relu function ReLU(x)=max{0,x} clearly is not a step-like activation function since it disobeys (a): $\sigma(1)=1$. A question naturally arises: is the set of all possible step-like activation functions large enough to cover up most situations (or further, is it possible to find the maximal set of activation functions that satisfies Theorem 3.4)?
  - In Definition 5.1, I assume in (b), $\sigma'$ is the derivative of $\sigma$. What does $\sigma^N$ in (c) mean? I searched over the paper but couldn't find an explanation.
  - To derive Corollary 3.5 from Theorem 3.4 it is necessary to show that sigmoid and tanh are special cases of step-like activation functions with $\eta=0$. However, Proposition 5.3 and Proposition 5.4 states that only a 3-layer NN with tanh and a 5-layer NN with sigmoid can be step-like activations. If this is the case, it largely restricts the structure of GNNs: the results in this paper are only applicable to MPNNs whose activations are again multi-layer NNs. This setting should be mentioned in the former parts of the paper when defining GNNs.
  - Continuing from the above concern. In Proposition 5.3 (and similarly in Proposition 5.4), a 3-layer NN is needed to express step-like activation function. Note that an activation function is a **point-wise** function that applies independently to each element of the input vector. How does the 3-layer NN behave like a poin-wise function to the input vectors in the whole GNN computation procedure? For example, let $X=[x]$ be the input and $W=[w_1,w_2]^T$ be the parameter at first layer. The output is $[w_1 x,w_2 x]^T$. If the dimension input is changed to be $X=[x_1,x_2]$, how does this reflect to $W$? To me this can be done by setting $W=[[w_1,w_2,0,0]^T,[0,0,w_1,w_2]^T]$ but this makes the number of parameters in each layer linerly grows w.r.t. the input dimension ( and thus makes the total number of parameters $O(d^2)$). Please correct me if I am wrong.

## Minor Issues:
- Corollary 5.5, 5.6, 5.7 all considers tanh activation. Should this be changed to arctan, tanh and sigmoid respectively?

## Suggestions:
- This paper can be further strengthened by discussing more complex GNN variants, e.g. higher-order GNNs. I suspect the number of parameters is logarithmic w.r.t. $N^{k-2}$ for $k$-order MPNNs, with $N$ being the number of nodes.

**Questions:**

Please refer to Weakness. There are some concerns that limit the impacts of the theory in this paper. I am willing to change the score if the concerns / problems are properly discussed / solved.

---

> ### Author Response · Authors · 2024-11-22
>
> Thank you a lot for taking the time to review our paper! We appreciate the comments that you raised about our paper.
>
> Regarding the weaknesses that you bring up, these are our answers:
>
> W1) It is true that ReLU is not a step-like activation function (according to our definition). In general, our proof of Theorem 3.4 relies on being able to approximate CReLU efficiently. However, we ignore the exact domain of applicability of Theorem 5.1. The question that you raise is an interesting question, but we are unable to provide a meaningful answer. However, we believe that step-like activation functions cover many of the cases with a monotone sufficiently smooth activation function—we hope that our examples showcase this.
>
> W2) You are right, $\sigma'$ stands for the derivative. Also, $\sigma^N$ stands for the composition of $\sigma$ with itself $N$ times. We have added this in the Notation section, as well as in other parts, to avoid confusion.
>
> W3 \& W4) Observe that if we can express an activation function $\tilde{\sigma}$ as an NN with activation function $\sigma$ (of course this NN has both a 1-dimensional domain and codomain as it is expressing a function $\tilde{\sigma}:\mathbb{R}\rightarrow\mathbb{R}$), then any NN with activation function $\tilde{\sigma}$ can be converted into an NN with activation function $\sigma$. Of course, this will affect the parameters of the resulting NN, but not in a significant way.
>
> Being more explicit, assume that $\tilde{\sigma}$ can be written as an NN with activation function $\sigma$ with $m$ layers of respective width $r_{\text{input}}=1,r_1,\ldots,r_{m},r_{\text{output}}=1$, where $r_{\text{input}}$ denotes the width of the input layer, $r_o$ the width of the output layer, and $r_1,\ldots,r_m$ the middle layers where the activation function $\sigma$ is applied. Then every NN with activation function $\tilde{\sigma}$ with $\ell$ layers $w_{\text{input}},w_1,\ldots,w_{\ell},w_{\text{output}}$, where $w_{\text{input}}$ is the dimension of the input vector and $w_{\text{output}}$ the dimension of the output vector, can be converted into a NN with activation function $\sigma$ with $\ell m$ layers
> $$
> w_{\text{input}},w_1r_1,\ldots,w_1r_m,w_2r_1,\ldots,w_2r_m,\ldots,w_{\ell}r_1,\ldots,w_{\ell}r_m,r_{\text{output}}.
> $$
> One can see that such an NN is not general having a lot of zero-weights and a so a very particular block structure. However, to show expressivity we only need to show that there is a NN that can express the function at hand.
>
> In our setting, the NN to express $\tilde{\sigma}$ in terms of $\sigma$ is a small fixed NN. In this way, using our definition of size of the NN (the sum of the widths), we can see that the size of the new NN
> $$
> w_{\text{input}}+(w_1+\cdots+w_{\ell})(r_1+\ldots+r_m)+w_{\text{output}}\leq (w_{\text{input}}+w_1+\cdots+w_{\ell}+w_{\text{output}})(1+r_1+\ldots+r_m+1)
> $$
> is bounded by the product of the sizes of the NN using $\tilde{\sigma}$ and the NN that expresses $\tilde{\sigma}$ using $\sigma$. Similar expression can be obtained for the number of parameters, but we hope that this clarifies why the combination of NNs that we do is permitted.
>
> Further, we added an appendix specifying this to make it clear in the paper.
>
> Minor Issues: We have corrected this. Thanks for pointing them out.
>
> Suggestion: We agree that this might be possible to do. However, we feel that this goes beyond the scope of the paper: to show that almost-uniform expressivity is possible for many activation function for which uniform expressivity fails. Even though we are certain the positive results from Section 5 would extend to higher order GNNs, we believe that there is not a simple extension of the impossibility results of Section 4.

---

> ### Comment · Reviewer_Mu1a · 2024-11-30
> **Response**
>
> Thank you for your response. I think the response has addressed most of my concerns.
>
> ### About W3 & W4
>
> ```
> However, to show expressivity we only need to show that there is a NN that can express the function at hand.
> ```
> This is true and I agree with you. Nevertheless, considering the major contribution of this paper is to show that MPNNs with step-like activations can express logic formulas without using **too many** parameters, I believe it is still necessary to explicitly discuss the number of parameters induced by similuting $\sigma$ functions with NNs. Therefore, the authors are encouraged to add relevent discussions in the final version. (I didn't see an Appendix section discussing about this; am I missing something?)
>
> Going on, I am still concerned about the correstness of Corollary 5.5, 5.6, 5.7. E.g., using your example, shouldn't the size of the GNNs in Corollary 5.5 have a term $O(d)$? Please correct me if I was wrong.
>
> Overall, I thank the authors for the response, and hope the above concerns could be clarified.

---

> > ### Author Response · Authors · 2024-12-01
> >
> > Thanks a lot for catching this error.
> >
> > Indeed, in Theorem 5.1, the bounds should have been respectively $\left(2+N+\frac{2+\log\Delta}{1-\log(1+\eta)}\right)d$ (we had a sign typo in the denominator, but this only affects the constants in our bounds) and $\left(2+N+\log(2+\log(\Delta+2)\right)d$. Therefore, as you pointed out the bounds in Corollaries 5.5, 5.6 and 5.7 are of the form $O(d\log \Delta)$ and $O(d\log\log\Delta)$. More concretely, the corrected bounds are $(9 + 3.5 \log(\Delta + 2))d$ for Corollary 5.5, and $(8+4\log(2+\log(\Delta + 2)))d$ for Corollaries 5.6 and 5.7.
> >
> > We will correct this for the final version of the paper.

---

> > > ### Comment · Reviewer_Mu1a · 2024-12-01
> > > **Response**
> > >
> > > I thank the authors for the response.
> > >
> > > After discussion, most of my concerns have been properly addressed. In the final version of the paper, the authors are supposed to:
> > > - Correct the upper bounds of the sizes of GNNs in theorems and corollaries.
> > > - Correct the corresponding proofs.
> > >
> > > Besides, I strongly encourage the authors to:
> > > - Discuss how simulating step-like activations by NNs with different activation function will affect the structure of NNs (as discussed above).
> > >
> > > I trust the authors to include them in the final version of the paper, and I believe the paper makes a nice contribution for understanding expressiveness measurements of GNNs. Thus, I have increased my score.

---

### Official Review · Reviewer_ikke · 2024-11-03

**Soundness:** 4
**Presentation:** 3
**Contribution:** 3
**Rating:** 8
**Confidence:** 3

**Summary:**

The paper aims to more closely characterize the uniform approximation capabilities of graph neural networks as a function of the activation functions applied to each aggregation of messages. There are primary theoretical contributions:
1. While previous results suggest that bounded-size guarded model logic (GC2) queries can be expressed by fixed-size GNNs with CReLU activations, Section 3.1 proves that certain GC2 circuits cannot be expressed by GNNs with sigmoid and tanh activations (or more generally, for superpolynomial Pfaffian activations). This limits the universality properties of GNNs. The proof involves constructing pairs of tree graphs whose outputs will be identical as a result of their activations.
2. However, Section 3.2 shows that all bounded-size GC2 queries can be expressed by a GNN whose size grows slowly as a function of the graph degree for sigmoid and tanh activations (and more generally, for "step-like activations").

They validate their theoretical results by training GNNs to learn two synthetic GC2 queries on different graph sizes. These experiments reveal sharp thresholds between the abilities of different activations to distinguish between similar graph instances.

**Strengths:**

The results of the paper are well-motivated theoretically---it is interesting to understand the trade-offs of different activations and the limitations of universal approximation. In particular, such fine-grained separations between constant-size and near-constant-size architectures are interesting, seeing as most graphs are small enough for the $\log\log\Delta$ scaling to be surmountable.

As far as I can tell, the mathematical results are correct. The proof ideas of distinguishing similar tree graphs and constructing separations with step-wise activations are intuitive.

**Weaknesses:**

In general, the presentation of the results would be improved by more context for Pfaffian and step-like activations. A more close linkage between tanh and sigmoid and the parameters of Pfaffian chain of order would appreciated, along with more examples of step-like activations and an intuitive discussion of their parameters.

While individually unimportant, there are several areas with ambiguous notation or minor mistakes that I would appreciate the authors clarifying:
* Definition 2.3 and Example 2.1 have some inconsistent notation, e.g. $E$ and $Edge$ are used interchangeably, and $\exists^{\geq p}$ is not defined (although it is fairly intuitive). The formula in Example 2.1 appears to indicate that at least one neighbor has at least two neighbors that are _red_, not blue.
* Definition 2.4 introduces a Pfaffian chain of order, but not Pfaffian functions. The relationship between the two should be made explicit.
* In Theorem 3.1, "Conversely, any (aggregation-combine) GNN expresses with activation function CReLU expressed a
GC2 query" does not parse.
* Grammar nit on line 250: "superpolynomial if it non constant".
* On line 291, what is the $y$ limit over? $y$ appears to be vector-valued.
* If Definition 5.1, does the $\sigma^N$ notation reference exponentiation or composition?

**Questions:**

In general, (approximate) linear threshold activations are not practical in neural networks because of lack of bounded derivatives. Are the authors concerned about the learnability of their constructions in Section 5 that simulate threshold activations?

What is the significance of the parameters $H$ and $\epsilon$ on the construction in Theorem 5.1? The other parameters seem to have influence on size complexity, but not those.

Likewise, do know whether the network weights for Theorem 5.1 can be bounded?

---

> ### Author Response · Authors · 2024-11-22
>
> Thanks for your detailed reading and appreciation of our work!
>
> Regarding the weaknesses that you raise:
>
> In answer to your comment about explicit connections between tanh, Sigmoid and their Pfaffian chain, we have added to the proof of Proposition 2.1 also an explicit chain for tanh. Regarding the parameters of step-like activation functions, we will a remark after Definition 5.1 or in the appendix if length does not permit.
>
> - We have corrected this in Example 2.1 and added a description of the interpretation of $E(x,y)$ in our context, which means that the variables $x$ and $y$ (as vertices) are connected in the considered graph.
>
> - Agreed. We have rewritten Definition 2.4. to be more explicit about what a Pfaffian function is, adding that is a function that is the last term $f_r$ of a Pfaffian chain.
>
> - Thank you. We have corrected this in the article.
>
> - Thank you. We have corrected this in the article.
>
> - We believe the answer to this question  should appear in the lines 292 and 293 just below.
>
> - This notations refers to composition, we have added an explanation of the notation in the paragraph ``Notations'' of Section 2.
>
> Regarding your three questions:
>
> - Indeed, we are concerned about it. Because of this, at the end of Section 6, we present some preliminary experimental results show the difficulty to actually learn GC2 queries. However, we plan a deeper study of learnability for future work.
>
> - Note that $N$, $H$ and $\varepsilon$ are interrelated due to (c) and (d) in Definition 5.1. For example, a large $H$ and a small $\varepsilon$ force a larger $N$. In this way, the parameters $H$ and $\epsilon$ influence the size of $N$ that is the one explicitly appearing in the complexity estimates that we give.
>
> - The GC2 query to be expressed will impact the amplitude of the weights. However, the parameters $\eta, \epsilon, N, H$ in the assumptions of Theorem 5.1 will not impact the weights amplitude, as we compose the step-like activation with itself. Only the speed of convergence to the step function, and so the number of layers, may depend on these parameters.

---

### Official Review · Reviewer_ir8P · 2024-11-04

**Soundness:** 3
**Presentation:** 2
**Contribution:** 3
**Rating:** 6
**Confidence:** 3

**Summary:**

This is a theoretical work about the connection between GC2 and GNNs with various activation functions. Its first contribution is to prove that uniform expressivity is not possible for GNNs with bounded Pfaffian activations. The second contribution is to build non-uniform expressivity for GNNs with Pfaffian activations via a constructive approach. Some synthetic experiments are provided to verify the theoretical arguments.

**Strengths:**

1. The motivation is good: rigorously understanding the expressive power of GNNs is a significant direction, especially it requires carefully quantify expressive capacities with sizes and depths. This work follows this motivation.
2. A rich family of activation functions is considered (such as arctan, tanh, sigmoid), as an extension of previous works about positive and negative results with simpler activation functions. And the general definition of Pfaffian functions makes it possible to generalize to other activations in the future.
3. The presented theory in the main text looks good.

Note that I didn't check all the proof in the appendix.

**Weaknesses:**

1. The presentation seems incomplete in some places, so thorough proofread and revision are necessary. For example:
    * in theorem 3.1, the last sentence is confusing in grammar.
    * in line 197, ''two blue neighbors'' should be ''red''.
    * in line 195, 199, 202, please check whether the parentheses are complete or not.
    * in line 279, the last sentence is not complete.
    * in Figure 2, $l_i$'s are missing. I suppose they are the leaf nodes.
    * around line 318, notations are not consistent, where line 685 uses $\nu_{t,m}(k)$ instead of $\nu_{t}(k,m)$
    * in line 363, what is the meaning of $\\sigma^N(R\\setminus (\\epsilon, 1-\\epsilon))$?
    * in Corollary 5.5, tanh -> arctan
    * in Figure 3, please use different linestyles for easiness of reading, instead of only using different colors
    * in line 495, a ''second'' shall be removed
2. around 335, the mentioned $T[0,k,m]$ and $T[1,k,m]$ seem to assume the tree in Figure 2 is rooted tree from $s$ without backtracking, which means $s$ is not in the leaf nodes. If so, please clarify this when defining the tree.
3. in line 495, please clarify how this layer argument is connected with theoretical results. Does it mean the different layers for activation functions in Prop 5.2-5.4?
4. in line 500, the conjecture about SGD is too sudden without evidence

**Questions:**

Please see the above concerns.

---

> ### Author Response · Authors · 2024-11-22
>
> Thank you for taking the time to read the paper! We appreciate your comments.
>
> We answer now the weaknesses that you are raising:
>
> 1. We have further proofread the paper and to eliminate existing typos and errors. In particular, we have corrected all the ones that you have pointed out. Thank you for such an extensive list.
>
> 2.  We have clarified that the trees $T[x,k,m]$ are rooted trees in $s$. Also,  note that a leaf node of a tree is defined as a vertex of degree $1$. So as soon as $m> 1$, $s$ is never a leaf of $T[x,k,m]$.
>
> 3.  In Section 6, we have now renamed the number of layers of the combine functions to $\ell$ to avoid confusion with the number of vertices at depth $2$ of the input graphs, $m$. The statement around line 495 was that the ability of the GNNs to separate both trees $T[0, k, m]$ and $T[1, k,m]$ is increasing as a function of $\ell$. We further proofread section 6 and made explanations clearer.
>
> 4. Thanks for pointing this out. We agree that the formulation was too sudden. We replaced this part as follows: ``The combination functions using step-like activation functions used in our approximation of GC2 queries have large gradients. We suspect this adds to the difficulty to efficiently train these networks to optimality, despite their ability to approximate queries to an arbitrary level of precision''.

---

### Author Response · Authors · 2024-12-03
**Global Response**

We thank the reviewers for their detailed assessment of our work, and their appreciation for the novelty and the potential impact of our paper. We are sincerely excited by the reviewers' reception that finds that our paper "studies GNN expressivity from a novel perspective" (Reviewer 6aa7) for "a rich family of activation functions" (Reviewer ir8P) "find[ing] a balance between general expressivity [...] and uniform expressivity [...] for practical needs" (Reviewer Mu1a) and "validat[ing] their theoretical results through experiment" (Reviewer ikke). We are grateful for all the comments and constructive feedback that have definitely helped to improve the overall quality of the paper.

For the final version, we will address all the comments of the reviewers and make numerous changes to improve the paper. Unfortunately, we missed the deadline to upload our reviewed version, so we specify all the significant prospective changes below. We note that despite these changes, the paper would remain essentially the same but with broader accessibility and readability.

In Section 2, we have simplified many of the definitions and expositions, making them more accessible. We have simplified Definition 2.1. regarding neural networks, we have restricted Definition 2.2. to the class of GNNs treated in our paper, we have rewritten Definitions 2.3 for GC2 queries, making it simpler and specifying precisely the depth; we have added a new definition to introduce the notation $Q(v,G)$ that was used before without definition; and we have simplified Definition 2.4 regarding Pfaffian functions. Additionally, we have added the meanings of $\sigma^N$ as composition power, $\sigma'$ as derivative and $\log$ as logarithm in base 2.

In Section 3, we have rewritten Definition 3.1. and Remark 3.1. for clarity. We further corrected the bounds of Theorem 3.4 and Corollary 3.5; see comments regarding Section 5 below.

In Section 4, we have added a description of the tree $T[x,m,k]$ within the text, as well as rearranged the update rule equations and corrected the formula $Q(v)$. We also added a comment to make clear that we are not restricted to one-dimensional vertex-embeddings.

In Section 5, we recall the notations $\sigma^N$ and $\sigma'$ before Definition 5.1. and we remark afterwards regarding the meaning of $N$, $\eta$, $H$ and $\varepsilon$. In Theorem 5.1, we have corrected, apart from an erroneous constant, two main errors in the bound: (1) The $d$ was not multiplying the two summands in the size expression. (2) The first bound should have been $1+\eta$ in the denominator and not $1-\eta$. In this way, the corrected bounds are respectively $\left(3+N+\frac{2+\log(\Delta+2)}{1-\log(1+\eta)}\right)d$ and $\left(3+N+\log\left(2+\log(\delta+2)\right)\right)$. Additionally, we have corrected the expressions depending on these expressions.

In Section 6, we have simplified the exposition of the numerical experiments to make it more accessible to the reader, making the exposition and notation more in line with the one used in the rest of the paper.

In Appendix A, we have rewritten Lemma A.2 to make it more comprehensive and added a lemma indicating the fact that a bounded Pfaffian function has a finite limit in $+\infty$.

In Appendix B, we have mainly corrected typos and clarified that the proof holds for GNNs producing vertex embeddings of arbitrary size.

In Appendix C, we have made some exposition changes. In C.1, we have rewritten the proof of Theorem 5.1. to make it more accessible and uniform between cases. In C.3, we have added Proposition C.4. indicating how the size of NN increases when we encode an activation function in terms of another NN.

In Appendix D, we have simplified the exposition.

---

### Meta-Review · Area_Chair_PKrS · 2024-12-26

**Metareview:**

This work  examines the connection between GC2 and GNNs with various activation functions, making two main contributions: proving the impossibility of uniform expressivity for GNNs with bounded Pfaffian activations, and demonstrating non-uniform expressivity through a constructive approach. The paper demonstrates solid theoretical foundations with well-motivated research direction, particularly in understanding GNN expressive power and the reviewers had a consensus that the paper has enough merits to be accepted.

**Additional Comments On Reviewer Discussion:**

The authors need to  make comprehensive revisions that address key reviewer concerns while maintaining the paper's core contributions. The improvements should mainly focus on: technical clarity, mathematical precision, and enhanced presentation. The authors have already outlined a clear set of modifications to be incorporated into the work during rebuttal.

---

### Decision · Program_Chairs · 2025-01-22

Accept (Poster)